# Manipulating Weyl quasiparticles by orbital-selective photoexcitation in WTe₂

Meng-Xue Guan [1,2,5], En Wang[1,2,5], Pei-Wei You [1,2], Jia-Tao Sun [3,1✉] & Sheng Meng [1,2,4✉]

Optical control of structural and electronic properties of Weyl semimetals allows development of switchable and dissipationless topological devices at the ultrafast scale. An unexpected orbital-selective photoexcitation in type-II Weyl material WTe₂ is reported under linearly polarized light (LPL), inducing striking transitions among several topologically-distinct phases mediated by effective electron-phonon couplings. The symmetry features of atomic orbitals comprising the Weyl bands result in asymmetric electronic transitions near the Weyl points, and in turn a switchable interlayer shear motion with respect to linear light polarization, when a near-infrared laser pulse is applied. Consequently, not only annihilation of Weyl quasiparticle pairs, but also increasing separation of Weyl points can be achieved, complementing existing experimental observations. In this work, we provide a new perspective on manipulating the Weyl node singularity and coherent control of electron and lattice quantum dynamics simultaneously.

[1] Beijing National Laboratory for Condensed Matter Physics and Institute of Physics, Chinese Academy of Sciences, Beijing 100190, China. [2] School of Physical Sciences, University of Chinese Academy of Sciences, Beijing 100190, China. [3] School of Information and Electronics, Beijing Institute of Technology, Beijing 100081, China. [4] Songshan Lake Materials Laboratory, Dongguan, Guangdong 523808, China. [5] These authors contributed equally: Meng-Xue Guan, En Wang. ✉email: jtsun@iphy.ac.cn; smeng@iphy.ac.cn

Three-dimensional Weyl semimetals (WSMs) are novel topological phases of matter, in which the chiral Weyl nodes can be viewed as pseudo-magnetic monopoles in momentum space and the magnetic charge is determined by chirality[1–7]. These magnetic monopoles have direct effects on the motion of electrons, providing an ideal platform to explore the nonlinear optoelectronic responses that related to topology in gapless materials[8,9]. The nonlinear responses of the WSMs, such as the second harmonic generation[10], nonlinear Hall effect[11–14], and topological phase transition[15–17], are of great importance in probing the fundamental properties of quantum materials as well as for applications in optoelectronic devices and solar cells.

Although a nonlinear optical process that involves high-energy excited states may not directly capture the low-energy singularity at the Weyl nodes, the ground state and low-energy excited states are characterized by the geometry and topological nature of the Bloch wavefunctions at the node points. For example, enhanced bulk photovoltaic effect (BPVE) is reported in two typical WSMs, i.e., type-I WSM TaAs[18] and type-II WSM TaIrTe$_4$[19], by mid-infrared light illumination. The BPVE is a nonlinear optical response that intrinsically converts linearly polarized light into electrical dc current through the shift of the charge center during the interband excitation[20–23]. The substantial enhancement of the generated photocurrent, i.e., shift current, is ascribed to the Berry flux field singularity of the Weyl nodes. The results indicate that the carrier excitation around the Weyl nodes display a range of unique behaviors, which are highly sensitive to the laser polarization, helicity, and wavelength.

The connection between macroscopic nonlinear optoelectronic responses and band topology of the WSMs are routinely established by models based on ground-state calculations and symmetry analysis[24,25]. However, there are two limitations: (i) The lack of the real-time dynamics of excited electrons in momentum-space and the corresponding real-space; (ii) The electron-lattice

and photon-lattice coupling are intrinsically ignored, which are of crucial importance in inducing topological phase transitions. Real-time time-dependent density functional theory molecular dynamics (TDDFT-MD) is a technique that may overcome these difficulties, and has been applied to unravel the interplay among different degrees of freedom in materials under photoexcitation[26–30].

In this work, we demonstrate that the photocurrent can be selectively switched by photoexcitation in bulk WTe$_2$, a type-II WSM, based on ab initio TDDFT-MD quantum simulations[31–33]. The current direction depends on the laser polarization as well as the photon energy, both of which play a vital role in determining the features of final excited states. The carriers can be excited around the Weyl nodes when a near-infrared laser pulse (photon energy $\hbar\omega = 0.5$–0.8 eV) is applied. Asymmetric excitation at the space-inversed **k**-points around the Weyl nodes is achieved, depending on linear light polarization and atomic orbital features of Weyl bands. In contrast, higher energy photons would excite electrons to high-energy bands, making the process polarization isotropic and of marginal relevance to the Weyl physics. The effective electron–phonon interactions drive switchable interlayer shear displacements, providing an ultrafast way for modulating topological properties, e.g., annihilation or increasing separation of Weyl nodes.

## Results

**Selective excitations under linearly polarized lasers.** The experimental geometry of bulk WTe$_2$ is adopted, which is characterized by an orthorhombic ($T_d$) unit cell without inversion symmetry[34]. The covalently bonded W atoms form a zigzag W-W chain along $a$-axis, leading to distinct anisotropy in the two-dimensional plane[35,36] (Fig. 1a). There are 8 Weyl points (WPs) in the $k_z = 0$ plane, and two of the WPs are shown in Fig. 2a,

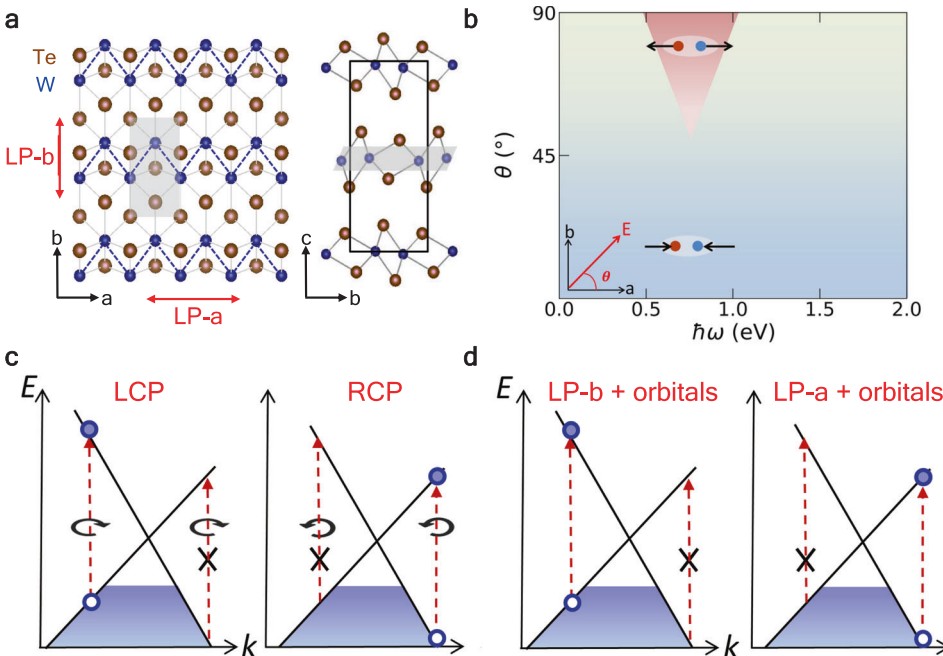

**Fig. 1 Illustration of selective photoexcitation in WTe$_2$. a** Lattice structure of $T_d$-WTe$_2$: top view ($a$-$b$ plane) and side view ($b$-$c$ plane), the dashed lines indicate the W-W zigzag chain along $a$-axis. The top layer is denoted by the grey shaded plane. **b** Phase diagram of laser-driven $T_d$-WTe$_2$ topological phase transition depending on photon energy $\hbar\omega$ and incident angle $\theta$ at the initial stages of the photoexcitation. The linearly polarized laser pulse propagates on the $a$-$b$ plane with the propagation direction as $\theta$. **c** Schematics of the chiral selection rule of the $\chi = +1$ Weyl node in momentum space under left- and right-handed circularly polarized (LCP, RCP) light. **d** Schematics of the orbital-dependent selective excitations when the laser pulses are linearly polarized along the $a$-axis (LP-a excitation) and $b$-axis (LP-b excitation).

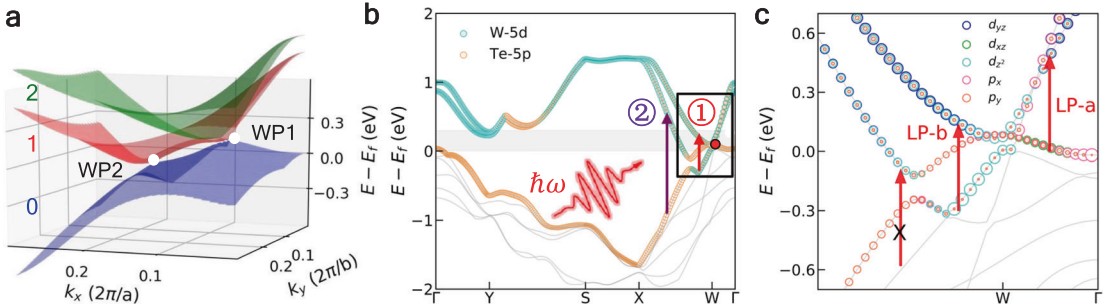

**Fig. 2 Band structure of WTe₂. a** Band structure of $T_d$-WTe₂ in the vicinity of two Weyl nodes. **b** Band structure along the high-symmetry lines in the Brillouin zone. The radiuses of cyan and yellow dots indicate the weight of W-5$d$ and Te-5$p$ orbitals. The red dot represents the position of WP1. Arrows ① and ② show the carrier transitions that near or far away from the Weyl node. The black rectangle represents the energy range where orbital-selective photoexcitation occurs. **c** The magnification of the band structure along X-Γ, and with more detailed atomic orbital information. The size of the colored dots represents the weight of the corresponding atomic orbitals. The red arrows represent the transitions with the laser linearly polarized along the $a$-axis (LP-a excitation) and $b$-axis (LP-b excitation) with a photon energy of 0.6 eV.

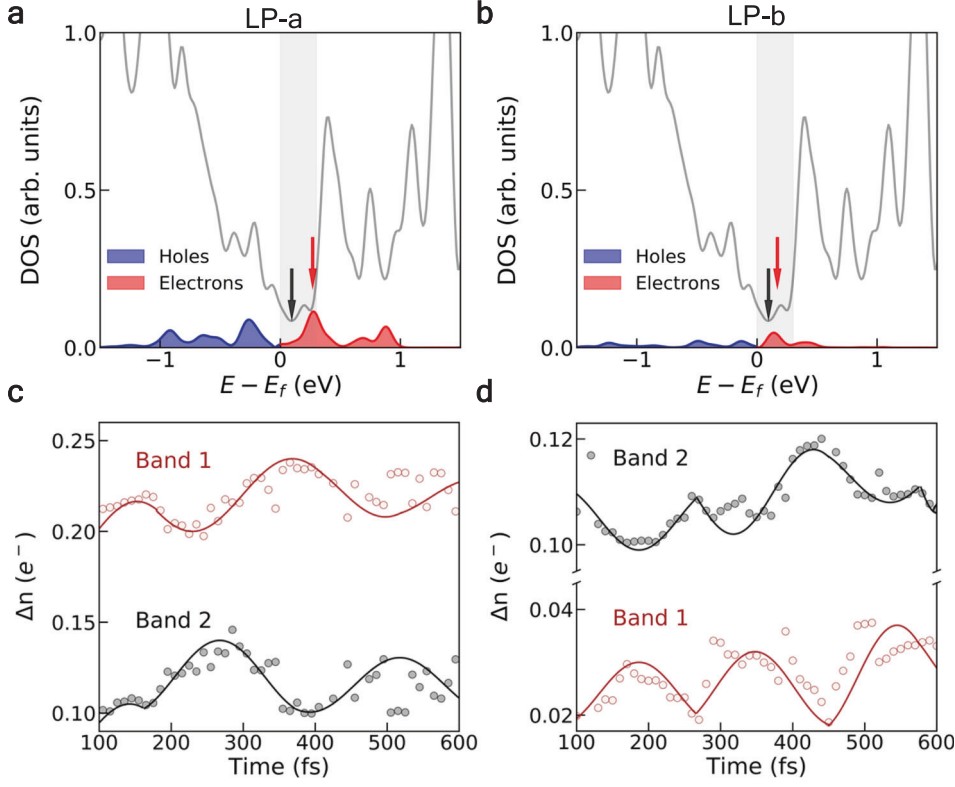

**Fig. 3 Carrier excitations in the momentum space. a** Energy distribution of the excited electrons and holes at $\hbar\omega = 0.6$ eV excitation linearly polarized along the $a$-axis (LP-a). The grey shaded area indicates the low-energy excitation region that near the Weyl node. The red and black arrows indicate the peak of the electronic excitation and the energy position of WP1, respectively. **c** Time evolution of the electronic occupation of band-1 (brown dots) and band-2 (grey dots) under LP-a excitation. The colored lines are guides to the eyes based on the oscillations of the electronic occupation. **b** and **d** are analogous to **a** and **c**, but are the results when laser linearly polarized along the $b$-axis (LP-b).

where the valence and conduction bands constituting the Weyl nodes are labeled as band-0 and band-1, respectively. Here, 'WP1' carries a Chern number $C = +1$ and locates nearly on the high-symmetry line Γ-X and is labeled as 'W' in Fig. 2b.

To study the optoelectronic responses of $T_d$-WTe₂, linearly polarized laser beams with time-dependent electric field $E(t) = E_0 cos(\omega t) exp[-(t - t_0)^2/2\sigma^2]$ are applied. The photon energy, width, and intensity are set as 0.6 eV, 10 fs, and $6.5 \times 10^{10}$ W/cm², respectively (Supplementary Fig. 1a). This setup allows us to reproduce a laser fluence (ca. 0.32 mJ/cm²) similar to experimental measurements[37]. There are 0.62% and 0.18% of valence electrons excited to the conduction bands after laser pulse ends

($t = 100$ fs), when the laser polarization is along the crystallographic $a$-axis (LP-a) and $b$-axis (LP-b), respectively (Supplementary Fig. 1b).

Figure 3a and b shows the energy distribution of excited electrons and holes after the laser pulse. It is clear that almost all electrons are promoted to low-energy excited states in the range of $E - E_f < 1$ eV ($E_f$ is the Fermi level) in both cases. Furthermore, a major part of electrons are promoted to states whose energy is less than 0.3 eV above $E_f$, the grey shaded region in Fig. 2b, indicating that these excitations involve transitions between the bands forming or nearing the Weyl cone. We find that there are two kinds of inter-band transitions: most of the electrons

are promoted to band-1 under LP-a excitation (Fig. 3c), while band-2 (band-2 is the adjacent, higher-energy conduction band) is dominant under LP-b excitation (Fig. 3d). Note that the tilted Weyl node leads to the partial occupation of band-1 along the W–X line in the Brillouin zone (BZ), where the band energies are below the $E_f$. The carrier transitions from the valence bands to the band-1 are thus forbidden along this direction due to the Pauli blocking effect. Electronic excitations take place at **k**-points along W–Γ (W–X) direction when the laser polarization is along a-axis (b-axis) (Fig. 2c). Therefore, optical transitions around the Weyl node can be controlled by manipulating laser polarization.

**Unique orbital features around the Weyl nodes.** The polarization controlled selective excitation arises from the asymmetric distribution of electron orbitals around the Weyl node. By projecting electronic wavefunctions of these bands onto atomic orbitals, one can clearly see that the dominant orbital components of band-0 and band-1 are exchanged due to the band inversion around the WP1 (Fig. 2b). According to the dipole selection rule, the transition probability depends on the geometry of the orbitals, i.e., $M_{if} = \langle \psi_f | \mathbf{A} \cdot \mathbf{p} | \psi_i \rangle = \langle \psi_f | \hat{S} | \psi_i \rangle$, where $M_{if}$ is the transition matrix element between the initial Bloch state $|\psi_i\rangle$ and final Bloch state $|\psi_f\rangle$, **p** is the electronic momentum operator, **A** is the electromagnetic vector potential and $\hat{S} = \mathbf{A} \cdot \mathbf{p}$. Based on group theory analysis, the integral of $M_{if}$ over all space will be zero if the direct product representation $\Gamma_f \otimes \Gamma_s \otimes \Gamma_i$ is not even parity under all symmetry operations of the group[38]. The space group of $T_d$-WTe$_2$ is $Pmn2_1(C_{2v})$, with two reflection symmetries (a mirror in the y-z plane and a glide plane in the x-z plane) and a two-fold rotation symmetry $C_2$ in the unitcell[6]. We find that $\langle d_{z^2} | S_x | p_x \rangle$ and $\langle d_{z^2} | S_x | d_{xz} \rangle$ are two dominant transition pathways along W-Γ under LP-a excitation, whereas $\langle d_{yz} | S_y | d_{z^2} \rangle$ is the most important pathway along W-X under LP-b excitation (Note crystallographic a-axis and b-axis correspond to the real-space x-axis and y-axis, respectively.) (Fig. 2c). These characteristic transitions are forbidden under the other excitation condition due to the broken of reflection symmetries, i.e., $\langle d_{z^2} | S_y | p_x \rangle = \langle d_{z^2} | S_y | d_{xz} \rangle = \langle d_{yz} | S_x | d_{z^2} \rangle = 0$. (for detailed analysis, see Supplementary Note 2). Therefore, in addition to regular spin selection rules around the Weyl points, the unique wavefunction features lead to a new kind of linear-polarization dependence of photoexcitation and selection of chirality.

We then discuss the similarities and discrepancies between the orbital-selective excitation and the well-known spin selection rules around the Weyl points[39,40]. Both of them uniquely determine the transition pathways within the Weyl cone. In the latter case, the absorption of a circularly polarized photon flips the spin, resulting in asymmetric excitations along the driving direction. For example, for a $\chi = +1$ Weyl fermion and a right-handed circularly polarized (RCP) light, the transition is allowed on the $+\mathbf{k}$ side but forbidden on the $-\mathbf{k}$ side of the Weyl node due to the conservation of angular momentum (Fig. 1c). The LP-a light in the present work, plays a similar role in introducing transitions as what RCP light do to ideal Weyl points (Fig. 1d). However, the selective excitation reported here is mutually determined by the linear polarization and atomic orbital features thanks to the combination of inversion symmetry breaking and finite tilts of the type-II Weyl dispersion, whereas, in spin selection rules, the coupling between the laser helicity and the chirality of the Weyl node is the dominant factor.

**Photocurrent from photoexcited Weyl semimetal.** The asymmetric carrier excitations at the Weyl points will contribute to

significant photocurrent generation[40,41]. Figure 4a, b shows the induced photocurrent along a-axis ($I_a$) and b-axis ($I_b$) under LP-a and LP-b excitations. Compared to the currents that are parallel to the incident polarization, the magnitudes of the perpendicular components are much smaller and show different oscillating behaviors. A nearly unidirectional current along the b-axis emerges with LP-a excitation, thanks to orbital symmetries and corresponding shift current in $T_d$-WTe$_2$. Detailed analysis of the microscopic mechanisms warrants further theoretical and experimental explorations, which is beyond the scope of this work.

The time integrals of $I_a$ and $I_b$ are shown in Fig. 4c, d, implying the accumulative effect of carrier excitation. After the laser pulse ends, the two polarizations along a-axis and b-axis induce net charges with the opposite directions, whose real-space distribution can be described by the charge density difference (CDD) (Fig. 4e, f). Here, time-dependent CDD is shown on a plane cutting along the top layer of the unit cell and is defined as $\Delta\rho(\mathbf{r},t) = \rho(\mathbf{r},t) - \rho(\mathbf{r},0)$, where $\rho(\mathbf{r},t)$ is the density at time $t$. The inhomogeneous carrier distribution will introduce a built-in electric field in the material and couple with specific phonon modes, providing possibilities to induce phase transitions.

**Switchable interlayer shear motion.** Ultrafast symmetry switches were recently observed in $T_d$-WTe$_2$ and MoTe$_2$, both accompanying with interlayer shear displacements along the b-axis[37,42]. Sie et al. predicted that the shear motion always starts from the $T_d$ phase to the centrosymmetric (1T' or 1T'(*)) phase regardless of polarization under the terahertz fields[37]. Most recently, Petra et al. demonstrated that for $T_d$-WTe$_2$, upon absorption of 827 nm laser pulses, not only the interlayer shear mode (~0.23 THz), but also higher frequency coherent phonon modes (2–6 THz) participate in the modulation of electronic structures during the phase transition[43]. The results indicate that electron-phonon interactions are of vital importance in controlling dynamic properties of $T_d$-WTe$_2$ on a timescale of picoseconds, which are closely related to the electronic transition pathways upon laser illumination[44].

The periodic oscillation in the population of photoexcited carriers helps to identify the presence of coherent phonon modes. In Supplementary Fig. 2, we show that consistent with the experiment observation, optical phonon modes at the frequency of ~4 THz are of particular relevance in the early stages of the phase transition[43]. A representative mode that belongs to the group of m-modes is shown in Supplementary Fig. 2c, where the atoms vibrate in the bc-mirror plane of the unit cell[45]. These high-frequency phonon modes have strong couplings to the low-frequency interlayer shear mode, which finally lead to transitions between the $T_d$ and the 1T' (or 1T'(*)) phases.

Here, we show that when the photon energy is specifically tuned, strongly anisotropic photoexcitaion and switchable interlayer displacements can be achieved. To describe the interlayer motion, averaged atomic displacement is calculated,

$$\Delta y(t) = \frac{1}{N} \sum_{i=1}^{N} \{y_i(t) - y_i(0)\}. \tag{1}$$

The summation runs over all atoms in the top (or bottom) layer in the unit cell, and $y_i(t)$ is the time-dependent position of atom $i$ along the b-axis.

We predict that the polarization-anisotropic response of interlayer shear displacement is achieved when a near-infrared laser pulse (ca. 0.6 eV) is applied, as shown in Fig. 5a and b. It is clear that the adjacent layers move with nearly identical velocity (≈0.25 Å/ps), but along opposite directions. With LP-a excitation, the bottom layer move towards the positive direction of the b-axis, while the top layer moves along the negative direction,

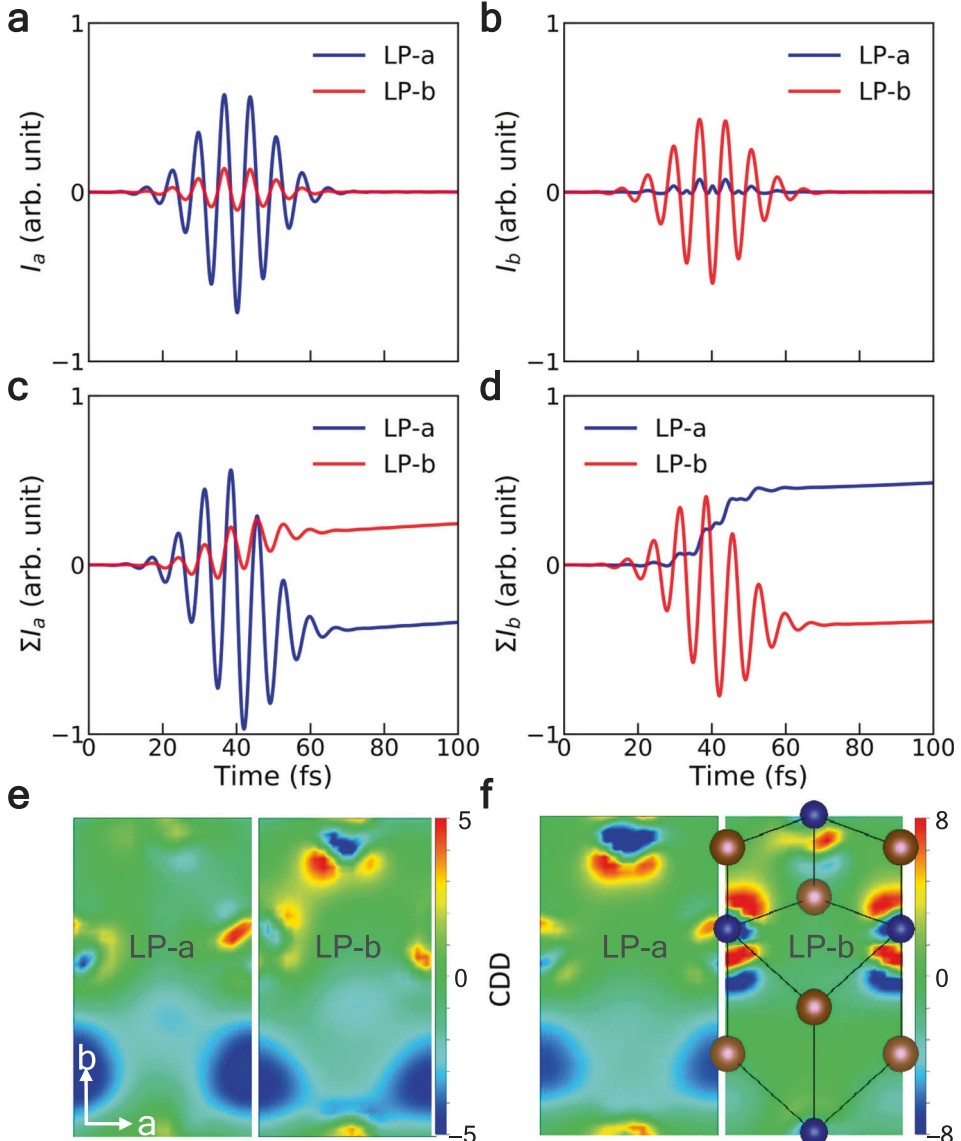

**Fig. 4 Laser polarization-dependent photocurrent. a, b** Laser-induced current along $a$-axis ($I_a$) and $b$-axis ($I_b$) under LP-a and LP-b excitations. **c, d** Time integrals of $I_a$ and $I_b$. **e, f** Charge density differences (CDD) between ground and excitation states at 50 fs (**e**) and 150 fs (**f**). The unit of CDD is $10^{-3} \times e/a_0^3$, $a_0$ is the Bohr radius. The red (blue) region represents the increase (decrease) in charge density.

leading to the restoring of inversion symmetry, in line with the experimental measurement[37]. However, the relative motion between the two layers is reversed under LP-b excitation, and the non-centrosymmetric order is further enhanced. Meanwhile, with the increase of laser amplitude, the atomic movements are accelerated and linearly dependent on the laser fluence (Supplementary Fig. 3). It implies that the shear motion can be switched towards either a centrosymmetric trivial phase or a non-centrosymmetric topological phase, accompanied by distinct evolution of potential energy surfaces.

The switchable shear motion is related to the singularity and chiral selection of the Weyl nodes, by orbital-selective excitation with the photon energy lying in the range of 0.5–0.8 eV (Supplementary Fig. 4). Similar methods have also been adopted to analyze the interlayer movement along $a$-axis and $c$-axis, however, negligible displacements are observed (Supplementary Fig. 5). It is ascribed to the strong electron-phonon coupling along $b$-axis due to the existence of effective interlayer-shearing phonon mode[45,46].

Limited by the unprecedented computational cost of first-principles excited-state dynamic simulations, the time duration we could simulate is less than 1 ps, which only describes the early stage of the photoinduced phase transition. It is reported that under moderate laser intensities, the interlayer displacements will oscillate with the frequency of the shear mode[37]. Qualitative prediction of the TDDFT lattice dynamics in a longer time scale can be achieved via comparing the time evolution of non-centrosymmetric degree (NCD) of WTe$_2$ between our results with experimental data. It can be verified using time-resolved second-harmonic generation (SHG) technique[37] or by monitoring the interlayer shear displacements (Supplementary Fig. 6). Figure 5c shows the normalized NCD when the pump fields are polarized at 45° off the crystallographic $a$-axis, for the initial $T_d$ phase, NCD is 1; for centrosymmetric phase, NCD is 0. The time-evolution of NCD matches well with the experimentally detected SHG signal, justifying the reliability of the present approach. In Supplementary Fig. 6d, we show that after $t = 600$ fs, the NCD of WTe$_2$ tends to return to its initial state under both LP-a and LP-b excitations.

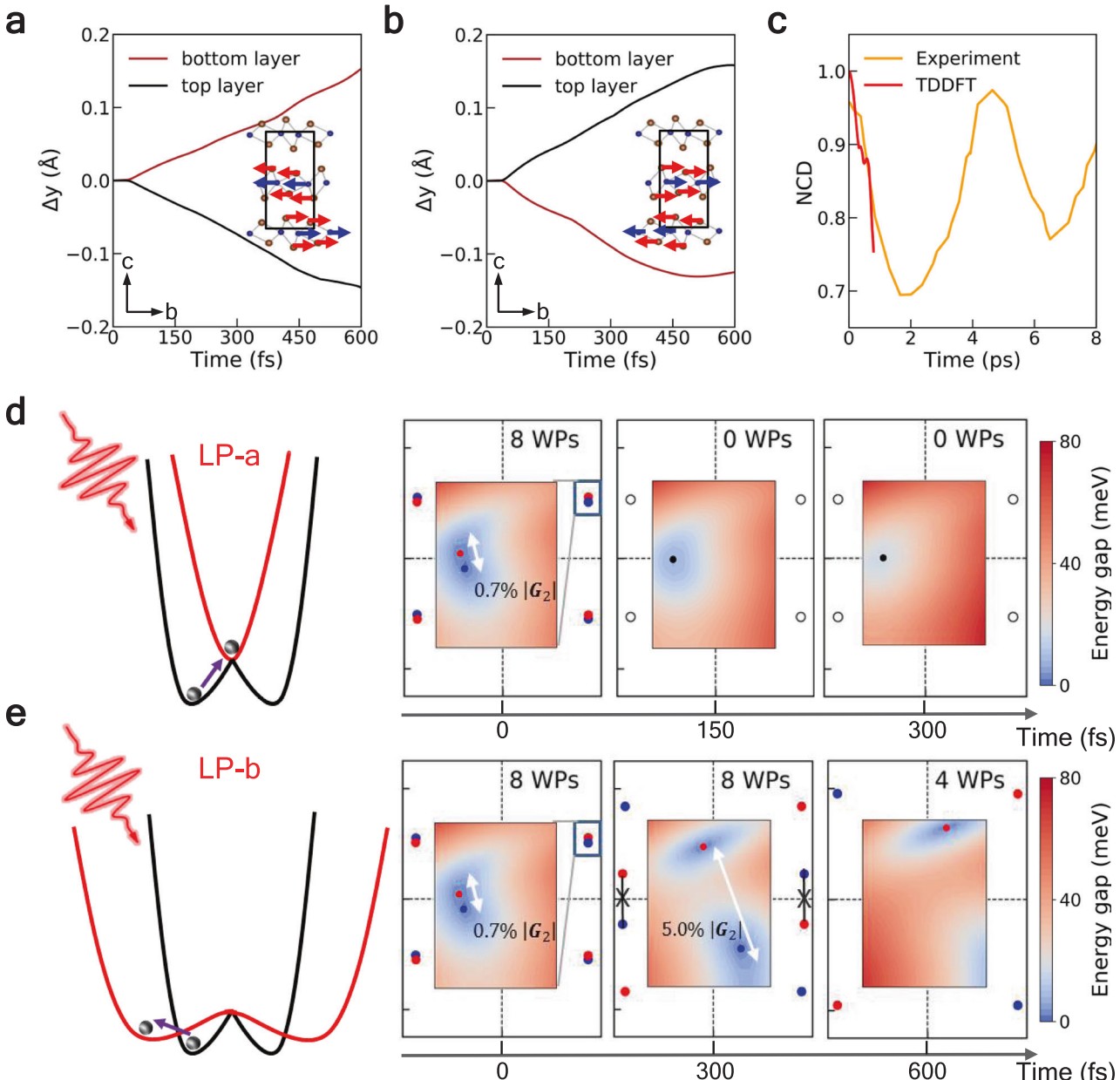

**Fig. 5 Polarization-anisotropic response of interlayer shear displacement and the induced topological phase transitions. a** Interlayer shear displacement under LP-a excitations. **b** Interlayer shear displacement under LP-b excitations. The inset in each panel shows the corresponding shear mode. **c** Comparison of non-centrosymmetric degree (NCD) between TDDFT result and the experimental detected SHG signal taken from ref. [37]. In both two cases, the pump fields are polarized at 45° off the crystallographic a-axis. The SHG time trace is measured at the laser pulse strength and wavelength are 3.1 MV/cm and 2.1 μm, respectively. **d**, **e** Schematic illustration of the potential energy surface before (black line) and after (red line) the photoexcitation, and time evolution of WPs under LP-a and LP-b excitations, respectively. The $k_z = 0$ plane ($-0.14|\mathbf{G_1}| \leq k_x \leq 0.14|\mathbf{G_1}|$, $-0.10|\mathbf{G_2}| \leq k_y \leq 0.10|\mathbf{G_2}|$, where $\mathbf{G_1}, \mathbf{G_2}$ is the lattice vector along x and y direction of the reciprocal space respectively) is shown. The inset in each panel shows the positions of nearest WPs.

Based on that, we expect the interlayer displacements might exhibit an oscillatory behavior in later stages.

**Light-controlled Weyl nodes**. The polarization-dependent shear motion provides an effective way to control topological properties of WTe$_2$. Depending on the direction of interlayer shearing, all WPs of opposite chirality will be annihilated under LP-a excitations, or separated further to have extremely long Fermi arcs under LP-b excitations. Figure 5d and e shows the time-dependent positions and numbers of WPs under the two excitation conditions. At equilibrium ($t = 0$ fs), the WPs are separated by 0.7% of the reciprocal lattice vector $|\mathbf{G_2}|$. Under LP-a excitation, the WPs move

towards each other and the annihilation occurs when $t = 150$ fs. Under LP-b excitation, the WPs are pushed away from each other in momentum space, leading to more robust, ideally separated WPs, e.g., the WPs are separated by 5% of $|\mathbf{G_2}|$ at $t = 300$ fs. Thus a large separation is crucial in realizing a giant quantum anomalous Hall effect, with Hall conductivity proportional to the separation between Weyl nodes[13,47,48]. Further increasing the displacement ($t = 600$ fs), a Lifshitz transition occurs from a topological semi-metal with eight WPs to the one with four WPs. Therefore, the number of the WPs can be modulated freely to zero, four, and eight by controlling the photon energy and the polarization of the laser pulse.

Selective excitation of WSMs emerge only if the excitation involves transitions between bands forming (or nearing) the Weyl cone, leading to polarization-dependent shear mode. However, when a terahertz laser pulse is applied, the photon energy (e.g., 23 THz ≈ 0.095 eV) is too low to promote electrons to higher-lying conduction bands (e.g., band-2), therefore, only intraband excitations and interband transitions between band-1 and band-0 (path 'LP-a' in Fig. 2c) is allowed regardless laser polarization. On the other hand, a much higher photon energy (e.g., 1.5 eV) would excite electrons to bands far above the Weyl points, making the process of marginal relevance to Weyl physics (path ② in Fig. 2b). Therefore, polarization-isotropic interlayer displacement is expected in the above two excitation conditions (i.e., THz and visible excitations), both lead to the annihilation of the WPs. In Supplementary Fig. 7 and Supplementary Fig. 8, we show that when the photon energy is 1.5 eV, the interlayer movement always starts from the $T_d$ phase to the centrosymmetric phase, which is consistent with experimental observations[37,42,43]. To show the distinct evolution of WPs separation, the phase diagram as functions of photon energy and incident direction is constructed, as shown in Fig. 1b.

## Discussion

The main results of this work established a connection between the unique orbital features around the Weyl nodes and the polarization-switchable interlayer displacement, which is significantly different from the works that appeared recently in literature. Most well-known papers on Weyl physics of WTe₂ do not present information on orbitals around the Weyl points;[6,49] a few latest articles indeed performed orbital analysis, however, only on bands far away from the Weyl points, thus are irrelevant to chiral physics[44]. Our investigation covers a wide range of photoexcitation conditions (photon energy: 0.5–1.5 eV; incident angle: 0–90°), and therefore the driving mechanisms, i.e., optical excitation and effective electron-phonon couplings are distinct from the THz-field-driven symmetry switch reported recently[37]. Most importantly, only restoring of inversion symmetry is presented in Ref. [37] where the pathway is speculative, we present here a complete phase diagram (Fig. 1b) where crystal asymmetry can be either destroyed or enhanced at wish, and the real-time dynamics is obtained fully from TDDFT quantum dynamics simulations.

Apart from the atomic movements, the topological properties of WTe₂ and related materials, i.e., MoTe₂ can be modulated via the direct laser-electron interactions. In the recent work, Beaulieu et al. proposed that the dynamical modification of the effective electronic correlations will lead to an ultrafast Lifshitz transition around the Fermi surface[50–52]. Another complementary picture of manipulating Weyl fermions is coherent Floquet driving, e.g., Hübener reported that a Floquet–Weyl semimetal is dynamically created by breaking time-reversal symmetry with circularly polarized lasers[53]. The above mechanisms are necessarily linked, and supplement each other (for more discussions, see Supplementary Note 10 to Note 12). Therefore, the real evolution of electronic structures should be the ensemble effect of different degrees of freedom, i.e., photon, electrons, and phonons, while in the present work, we focus our attention on the lattice dynamics induced topological phase transitions.

Based on our simulations, the switchable interlayer displacements and the induced WPs evolution will establish an oscillatory fashion around their equilibrium positions, as shown in Fig. 5c. The experimental observation of the above dynamics should be transient and locked-in to the frequency of the sheer mode, and therefore, ultrafast techniques with high temporal resolution (~300 fs) are needed. Apart from ultrafast pump-probe and time-resolved second-harmonic-generation (SHG) or electron diffraction (UED) spectroscopies[37,42], one potential candidate is time-

and angle-resolved photoemission spectroscopy (trARPES). It has a demonstrated advantage in investigating the structure of arcs, connectivity of electron and hole pockets, and WPs positions. However, direct measurement of the dynamic WPs separation requires sufficient energy and angular resolution to distinguish the terminal of Fermi arc[43,54,55], which might be the major challenges for the experimental observation.

In summary, our ab initio TDDFT-MD simulations reveal that the phase transition in type-II WSM WTe₂ can be controlled by orbital-selective optical excitations, mediated by effective electron-phonon couplings. Polarization-switchable interlayer displacement is predicted when the electronic transitions are in the vicinity of the Weyl cone, where orbital-dependent excitations of WSMs emerge with a suitable photon energy. In this scenario, topological phase transitions can be controlled towards either annihilating all WPs or inducing largely separated WPs. However, polarization-isotropic response of shear motion is expected if the transitions are far away from the Weyl cone, inducing only WPs annihilation. Our work provides a new insight on controlling Berry flux field singularity around the Weyl nodes, and the ab initio approach adopted here might be useful for understanding a wide range of non-linear responses of topological materials.

## Methods

**TDDFT calculations**. The TDDFT-MD calculations are performed using the time dependent ab initio package (TDAP)[31–33]. The bulk WTe₂ in its $T_d$ phase is simulated with a unit cell of 12 atoms with periodical boundary conditions. Numerical atomic orbitals with double zeta polarization (DZP) are employed as the basis set. The electron-nuclear interactions are described by Troullier-Martins pseudopotentials, PBE functional[56]. An auxiliary real-space grid equivalent to a plane-wave cutoff of 250 Ry is adopted. To make a good balance between the calculation precision and cost, a Γ-centered 6 × 5 × 3 **k**-point grid is used to sample the Brillouin zone. The coupling between atomic and electronic motions is governed by the Ehrenfest approximation[57]. During dynamic simulations the evolving time step is set to 0.05 fs for both electrons and ions in a micro-canonical ensemble.

**Topological property calculations**. DFT calculations of the electronic band structure of WTe₂ at different interlayer displacement Δy (Fig. 5c and d) are performed by Vienna ab initio simulation package (VASP 5.4)[58]. Exchange-correlation effects are treated at the level of the generalized gradient approximation (GGA) through the PBE functional. The Projector-augmented wave (PAW) potentials with valence electronic configurations of {$6s^2$, $5d^4$} for W and {$5s^2$, $5p^4$} for Te are employed in conjunction with a plane-wave energy cutoff parameter of 300 eV. For self-consistent electron density convergence, a Γ-centered 12 × 10 × 6 **k**-point grid and Gaussian smearing with a smearing parameter of 0.05 eV are used. To obtain the positions of the Weyl Points (WPs) during the ultrafast topological phase transition, Wannier90[59,60] interface and WannierTools[61] are used to investigate the topological properties.

## Data availability

The data that support the plots within this paper and other findings of this study are available from the corresponding authors upon reasonable request.

## Code availability

The code and mathematical algorithms that support the findings of this study are available from the corresponding authors upon reasonable request.

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

## Acknowledgements

We acknowledge partial financial support from the National Natural Science Foundation of China (Nos. 12025407, 91850120, 11774396, 11934003, and 11974045), National Key Research and Development Program of China (Nos. 2016YFA0300902, 2015CB921001, 2016YFA0202300, and 2020YFA0308800), and "Strategic Priority Research Program (B)" of Chinese Academy of Sciences (Grant Nos. XDB330301 and XDB30000000). We acknowledge helpful discussions with Prof. Shengbai Zhang.

## Author contributions

S.M. conceived, designed, and supervised the research. Most of the calculations were performed by M.G. and E.W., with the help from P.Y. and J.S. All authors contributed to the analysis and discussion of the data. M.G., E.W., and S.M. wrote the manuscript. M.G. and E.W. contributed equally to this work.

## Competing interests

The authors declare no competing interests.
