## [Peer Review File · Nature Communications]

Reviewers' Comments:

Reviewer #1:

Remarks to the Author:

The authors provide a theoretical study of possible light-induced changes to the electronic structure in the Weyl material Td-WTe₂. As their method of choice they use time-dependent density functional theory (TDDFT) combined with molecular dynamics (MD) to capture the adiabatic effects of ionic motion triggered by a short laser pulse (photon energy between 0.5 and 0.8 eV).

I have some general questions about the validity of the chosen approach, and would like the authors to consider and answer the following questions:

(1) In a recent experimental study on the closely related material Td-MoTe₂, it was pointed out that the role of Hubbard U and its dynamics after photoexcitation can be important for the obtained band structures: S. Beaulieu et al., <https://arxiv.org/abs/2003.04059>

Here the authors employ the PBE functional, which should in principle capture some of these correlation effects, but it would be clearer if a direct comparison was included, e.g., with LDA+U as discussed here: D. De Sante et al., <https://journals.aps.org/prl/abstract/10.1103/PhysRevLett.119.026403>

Does the PBE functional allow for time-dependence of the correlations (dynamical screening effects)? Since correlation effects are similar in these two compounds, I would expect a similar dynamics to be important on short to intermediate time scales, which are relevant here.

(2) It would be good if the authors could compare and contrast their approach to the complementary picture of manipulating Weyl fermions by coherent Floquet driving, see, e.g., H. Hübener et al., <https://www.nature.com/articles/ncomms13940>

If I understand correctly in their approach the coherence of the laser does not play any role, and the main purpose of the photoexcitation is to trigger an adiabatic lattice dynamics that then leads to effects on the electronic structure.

(3) As a follow-up question relating to points (1) and (2) above: The photoexcitation density here is investigated by projecting the excited state wavefunction onto the groundstate orbitals (Eq. S4). In the work by Beaulieu et al. above it was pointed out that the dynamics of the wavefunctions and change of electronic structure due to photoexcitation played a key role in their experiment, even before lattice effects become important. I would find it relevant to discuss this issue as the authors here seem to rely entirely on the lattice dynamics to induce the proposed effects.

In summary, I think these questions should be properly addressed before I can provide a definitive opinion on the manuscript.

Reviewer #2:

Remarks to the Author:

The manuscript contains a theoretical analysis of the photoexcitation of the material WTe₂. The focus is on the pump-field polarization dependence of the photoexcited dynamics of the material.

It is showed that photoexcitation with a 0.6eV pulse polarized along the (a) and (b) cristallographics axis

leads to the excitation of the interlayer shear motion in opposite directions.

It is argued that this polarization dependence can be used to tune the position and the number of the Weyl Points (WPs) in the Brillouin Zone.

It is therefore proposed that the light-polarization could be used to tune the topological properties of this material.

The idea of using light to control topological properties (namely the position and the number of the WPs) using light stimulation of the interlayer shear mode in WTe₂ is not new.

In fact, it has been recently investigated experimentally in Nature 565, 345 (2019) [Ref.37] which is used by the authors to compare part of their theoretical results.

The authors provides an extended microscopic description of this phenomenon based on an ab-initio description of the photoexcited dynamics.

This study makes the additional observation that, based on a symmetry analysis of the bands close to the WPs, this phenomenon can be controlled using the polarization of the incident light.

The polarization dependence of the photoexcitation in WTe₂ is an already documented fact, see for example ARPES data

in J. Phys. Condens. Matter 32, 345503 (2020) [Ref. 40].

This paper contains an interesting observation which builds on the combination of the two aspects above.

Overall I think that this represents an incremental advancement of the knowledge about the light-control of topological properties of WTe₂.

The paper seems scientifically sounds, even if there are points that needs to be addressed in more detail.

I find that the symmetry analysis which is at the basis of the polarization selection rules is not sufficiently explained.

Moreover, except for the plot in Fig.5c, the paper lacks of an in-depth comparison with the experimental observation of Ref. 37.

In Ref. 37 the photoinduced shear mode was explicitly reported to be polarization independent (see Fig. 3c in ref. 37).

Since the present study points towards a polarization dependence of the shear mode excitation, this aspect needs to be discussed.

The paper can be certainly improved, but my opinion is that this work does not meet the criteria for publication of Nature Communication.

I append here a list of comments for the authors.

*) The symmetry analysis of the selection rules is not sufficiently explained.

Bands 0 1 and 2 that seems to play a major role in this. These bands are shown in Fig. 2a, but I find it difficult

to recognize these bands in panels b and c where many more bands are plotted.

Next to that the selction rules related to the dipole matrix elements $M_{ij} = \langle \Psi_i | X | \Psi_j \rangle$.

What are the crystal symmetries that leads to the vanishing of cerntains matrix elements?

For example, similar aspects has been recently addressed in the context of the excitonic insulator condidate

Ta₂NiSe₅ (see PRR 2, 013236 (2020) and PRL 124, 197601 (2020)).

Also, what is the meaning of the states Ψ_i ? Are these Bloch states, or Wannier projection or atomic orbitals?

In the latter case, selection rules usually refer to spherical symmetry. How these are extended to the solid?

*) In Ref. 37 the excitation of the interlayer shear mode was first achieved with a pump in the THz regime.

What is expected in the theory analysis for a frequency smaller than the 0.6 eV considered here? Also, what is the expected fluence dependence?

*) What is the polarization used in the experimental data used to compare the TDDFT results at LP-polarization?

Comments from Reviewer 1:

1. Overview

The authors provide a theoretical study of possible light-induced changes to the electronic structure in the Weyl material Td-WTe₂. As their method of choice they use time-dependent density functional theory (TDDFT) combined with molecular dynamics (MD) to capture the adiabatic effects of ionic motion triggered by a short laser pulse (photon energy between 0.5 and 0.8 eV).

I have some general questions about the validity of the chosen approach, and would like the authors to consider and answer the following questions:

Our response: We thank the reviewer for the pertinent and constructive suggestions. We have addressed all concerns of the reviewer and revised the manuscript thoroughly.

Comments 1: *In a recent experimental study on the closely related material Td-MoTe₂, it was pointed out that the role of Hubbard U and its dynamics after photoexcitation can be important for the obtained band structures: S. Beaulieu et al., <https://arxiv.org/abs/2003.04059>. Here the authors employ the PBE functional, which should in principle capture some of these correlation effects, but it would be clearer if a direct comparison was included, e.g., with LDA+U as discussed here: D. De Sante et al., <https://journals.aps.org/prl/abstract/10.1103/PhysRevLett.119.026403>. Does the PBE functional allow for time-dependence of the correlations (dynamical screening effects)? Since correlation effects are similar in these two compounds, I would expect a similar dynamics to be important on short to intermediate time scales, which are relevant here.*

Our response:

We thank the reviewer for the suggestion. We fully agree with the reviewer that the LDA/GGA functionals tend to over-delocalize electrons and underestimate the correlation effects (or dynamical screening effects). The DFT/TDDFT+U approach is

an effective method that might help to improve the accuracy of theoretical description of the strongly correlated systems. The modification of band structure induced by correlation effect [PhysRevLett.119.026403(2017)] and the ultrafast Lifshitz transition [arXiv:2003.04059] are important in understanding the physics of band topology.

To examine the effects of electron correlation in the present work, the band structure of T_d -WTe₂ is now calculated based on both DFT and DFT+ U approaches. Figure R1 shows the atomic-orbital projected band structure with $U=0$ (standard PBE calculation) and $U=2$ eV. The atomic orbitals around the Weyl node and their symmetry characteristics, which are of vital importance in the present work, are nearly identical with different Hubbard U . Based on this fact, we think that the correlation effects would not alter the major conclusion in this work, i.e., a switchable interlayer shear motion with respect to light polarization.

Figure R1. PBE+ U band structures of T_d -WTe₂ around the Weyl node for $U=0$ and $U=2$ eV along the k_x direction.

We also note that the main point addressed in this work is the topologically phase transition induced by the lattice dynamics, where the correlation effect, albeit weak, might be kept constant during the phase transition. This mechanism is essentially different to correlation-driven effects, such as those discussed in [arXiv:2003.04059]. We believe that our simulation unravels real electronic dynamics, e.g., the concurrent appearance of the γ -pockets on the Fermi surface and the Weyl node annihilation.

The following discussions and relevant references have been added to the revised manuscript on

1) Supplementary Information (SI), Page 11:

“Supplementary Note 10: Electron correlation effects

The LDA or GGA functionals tend to over-delocalize electrons and underestimate the correlation effects. The DFT/TDDFT+ U approach is an effective method that might help to improve the accuracy of theoretical description of the strongly correlated systems⁵⁰. The correlation effect induced modification of band structure⁵¹ and the ultrafast Lifshitz transition⁵² are important knowledge in understanding the topological physics. However, the main points that addressed in this work is the topologically phase transition induced by the lattice dynamics, which can be controlled via orbital-selective photoexcitation. The above two mechanisms are not the same thing, but they are necessarily linked, and complementary. We believe that the real electronic dynamics is the combination of the two pictures.

To demonstrate that the correlation effect plays a minor role in our work, band structure of T_d -WTe₂ is calculated based on DFT+ U approach. Figure S9 shows the atomic-orbital projected band structure with $U=0$ (standard PBE calculation) and $U=2$ eV. In our work, the atomic orbitals around the Weyl node and their symmetry characteristics are of vital importance, which nearly identical with different Hubbard U . Based on that, we proposed that the correlation effect will not influence the main conclusions in this work, i.e., a switchable interlayer shear motion with respect to linear light polarization.

Fig. S9 | PBE+ U band structures of T_d -WTe₂ around the Weyl node. **a**, $U=0$ and **b**, $U=2$ eV. ”

2) New references are added:

“50. Tancogne-Dejean N, Oliveira MJT, Rubio A. Self-consistent DFT+ U method for real-space time-dependent density functional theory calculations. *Phys Rev B* 2017, 96(24): 245133.

51. Di Sante D, Das PK, Bigi C, Ergonen Z, Gurtler N, Krieger JA, *et al.* Three-Dimensional Electronic Structure of the Type-II Weyl Semimetal WTe₂. *Phys Rev Lett* 2017, 119(2): 026403.

52. Ultrafast Light-Induced Lifshitz Transition. *arXiv:200304059.*”

Comments 2: *It would be good if the authors could compare and contrast their approach to the complementary picture of manipulating Weyl fermions by coherent Floquet driving, see, e.g., H. Hübener et al., <https://www.nature.com/articles/ncomms13940>. If I understand correctly in their approach the coherence of the laser does not play any role, and the main purpose of the photoexcitation is to trigger an adiabatic lattice dynamics that then leads to effects on the electronic structure.*

Our response:

We are grateful for the reviewer’s suggestion of comparing and contrasting our

approach to the complementary picture of manipulating Weyl fermions by coherent Floquet driving.

In the present work, we focus on laser induced lattice dynamics, which results in a lattice-mediated topological phase transition. As the referee correctly pointed out, the laser can also modify the electron band structure in the Floquet picture through the direct laser-electron interaction. The following analysis based on Floquet theory indeed suggest that the linearly polarized laser will not change the Weyl point positions, only leading to optoelectronic band replicas, as explained below.

For comparison, Hübener *et al.* proposed that Weyl fermions can be modulated via the direct coupling between the laser field and electrons. In their groundbreaking work, circularly polarized lights are applied to split the Dirac point of Na₃Bi into two Weyl points by introducing the time reversal symmetry (TRS) breaking. While for *T_d*-WTe₂ studied in the present work, the robust Weyl quasiparticles already exist due to the intrinsic absence of inversion symmetry.

We find that the Floquet driving effect from linearly polarized laser is negligible in WTe₂. Let us consider the model Hamiltonian [Nature **527**, 495 (2015)]

$$H_0(\mathbf{k}) = Ck_x + \mathbf{k} \cdot \boldsymbol{\sigma}, \quad (\text{R1})$$

which realize a low-energy effective Weyl point with a +1 Chern number. The parameter C can adjust the type of this Weyl point and $\boldsymbol{\sigma} = (\sigma_x, \sigma_y, \sigma_z)$ is the Pauli matrix vector. Without loss of generality, we set the vector potential of laser as $\mathbf{A}(t) = A_0(\cos\Omega t, 0, 0)$. The time-dependent Hamiltonian $H(\mathbf{k}, t)$ is introduced by Peierls substitution

$$\begin{aligned} H(\mathbf{k}, t) &= H_0(\mathbf{k} - \mathbf{A}(t)) \\ &= C(k_x - A_0\cos\Omega t) + (k_x - A_0\cos\Omega t)\sigma_x + k_y\sigma_y + k_z\sigma_z. \end{aligned} \quad (\text{R2})$$

We construct the effective Floquet Hamiltonian perturbatively from the Fourier decomposition of $H(\mathbf{k}, t)$,

$$H_0 = H_0(\mathbf{k}), \quad (\text{R3})$$

$$H_{\pm 1} = -CA_0 - A_0\sigma_x, \quad (\text{R4})$$

where $H_n = \frac{1}{T} \int_0^T H(\mathbf{k}, t) e^{it\Omega n} dt$. Since the commuter of H_{-1} and H_1 equals to 0,

the effective Floquet Hamiltonian is

$$H_{eff} = H_0 + \frac{[H_{-1}, H_1]}{\Omega} + \mathcal{O}(A_0^4) = H_0(\mathbf{k}) + \mathcal{O}(A_0^4), \quad (\text{R5})$$

which retains the characteristics of the original band structure. The linear polarized light thus keeps the Weyl point positions unchanged. The only effect caused by direct laser-electron interaction might be the presence of Floquet band replicas due to the virtual processes of photon absorption and emission.

The following discussions and relevant references have been added to the revised manuscript on

1) SI, Page 12:

“Supplementary Note 11: Floquet driving effects

Based on the Floquet picture, the electronic band structure can also be modulated through the direct laser-electron interactions. For example, Hübener *et al* demonstrated that Floquet-Weyl point trajectories can be controlled in Na₃Bi via applying two time-delayed circularly polarized laser pulses⁵³. In their work, circularly polarized lights are used to split the Dirac fermion into two Weyl fermions by introducing the time reversal symmetry breaking, and the tunable delay time leads to varied field strength. While in *T_d*-WTe₂, there is an intrinsic inversion breaking and the induced Weyl fermions. Based on our simulations, linearly polarized laser will not change the Weyl point positions but only lead to some replicated bands.

Let us consider the model Hamiltonian⁶

$$H_0(\mathbf{k}) = Ck_x + \mathbf{k} \cdot \boldsymbol{\sigma}, \quad (\text{S9})$$

which realize a low-energy effective Weyl point with a +1 Chern number. The parameter *C* can adjust the type of this Weyl point and $\boldsymbol{\sigma} = (\sigma_x, \sigma_y, \sigma_z)$ is the Pauli matrix vector. Without loss of generality, we set the vector potential of laser as $A(t) = A_0(\cos\Omega t, 0, 0)$. The time-dependent Hamiltonian $H(\mathbf{k}, t)$ is introduced by Peierls substitution

$$\begin{aligned}
H(\mathbf{k}, t) &= H_0(\mathbf{k} - \mathbf{A}(t)) \\
&= C(k_x - A_0 \cos \Omega t) + (k_x - A_0 \cos \Omega t)\sigma_x + k_y \sigma_y + k_z \sigma_z. \quad (\text{S10})
\end{aligned}$$

We construct the effective Floquet Hamiltonian perturbatively from the Fourier decomposition of $H(\mathbf{k}, t)$

$$H_0 = H_0(\mathbf{k}), \quad (\text{S11})$$

$$H_{\pm 1} = -CA_0 - A_0 \sigma_x, \quad (\text{S12})$$

where $H_n = \frac{1}{T} \int_0^T H(\mathbf{k}, t) e^{it\Omega n} dt$. Since the commutator of H_{-1} and H_1 equals to 0, the effective Floquet Hamiltonian is

$$H_{eff} = H_0 + \frac{[H_{-1}, H_1]}{\Omega} + \mathcal{O}(A_0^4) = H_0(\mathbf{k}) + \mathcal{O}(A_0^4), \quad (\text{S13})$$

which retains the characteristics of the original band structure. The linear polarized light thus keeps the Weyl point positions unchanged. The only effect caused by direct laser-electron interaction might be the presence of Floquet band replicas due to the virtual processes of photon absorption and emission.”

2) New reference is added:

“53. Hubener H, Sentef MA, De Giovannini U, Kemper AF, Rubio A. Creating stable Floquet-Weyl semimetals by laser-driving of 3D Dirac materials. *Nat Commun* 2017, **8**: 13940.”

Comments 3: As a follow-up question relating to points (1) and (2) above: The photoexcitation density here is investigated by projecting the excited state wavefunction onto the groundstate orbitals (Eq. S4). In the work by Beaulieu et al. above it was pointed out that the dynamics of the wavefunctions and change of electronic structure due to photoexcitation played a key role in their experiment, even before lattice effects become important. I would find it relevant to discuss this issue as the authors here seem to rely entirely on the lattice dynamics to induce the proposed effects.

Our response:

We thank the reviewer for the comment. The direct impact of the electronic dynamics due to photoexcitation can be investigated by monitoring the time-evolution of electronic population in the momentum space with a fixed atomic geometry, as shown in Fig. R2. As discussed in the reply to comments 1 and 2, the Weyl node separation (or annihilation) and the dynamics of the wavefunctions are indirectly linked via the atomic displacements.

Figure R2 (a-c) shows the difference in photocarrier occupation numbers between the ground state ($n(0 \text{ fs})$) and the excited states ($n(t)$) during laser illumination with the fixed atomic geometry. It is obvious that at the end of laser pulse, most of excited carriers are relaxed to energy region around the Weyl node. After the end of laser pulse ($t = 70 \text{ fs}$), the intraband carrier scattering is the dominant interaction in the system (Fig. R2d), which might play an important role in modulating the carrier-phonon couplings and the topological phase transition. Therefore, the laser-induced real-time evolution of the carrier-carrier and carrier-phonon couplings are intrinsically coupled in our approach, while we mostly focus on the modulation effect of lattice dynamics in the present work.

Figure R2. Dynamics of photocarriers in the reciprocal space with the fixed atomic geometry. The red (blue) dots represent the increase (decrease) of electronic occupation.

In the revised manuscript, we added the following discussions and relevant

references to clarify the validity of the chosen approach and the novelty of this work.

1) On Page 9, we add

“Apart from the atomic movements, the topological properties of WTe₂ and related materials, i.e., MoTe₂ can be modulated via the direct laser-electron interactions. In the recent work, Beaulieu *et al.* proposed that the dynamical modification of the effective electronic correlations will lead to an ultrafast Lifshitz transition around the Fermi surface^{50, 51, 52}. Another complementary picture of manipulating Weyl fermions is coherent Floquet driving, e.g., Hübener reported that a Floquet–Weyl semimetal is dynamically created by breaking time-reversal symmetry with circularly polarized lasers⁵³. The above mechanisms are necessarily linked, and supplement for each other (for more discussions, see Supplementary materials). Therefore, the real evolution of electronic structures should be the ensemble effect of different degrees of freedom, i.e., photon, electrons and phonons, while in the present work, we focus our attention on the lattice dynamics induced topological phase transitions.”

2) In supporting information (SI, Page 14), Fig.R2 and related discussions have been added.

3) New references are added:

“50. Tancogne-Dejean N, Oliveira MJT, Rubio A. Self-consistent DFT+U method for real-space time-dependent density functional theory calculations. *Phys Rev B* 2017, **96**(24): 245133.

51. Di Sante D, Das PK, Bigi C, Ergonen Z, Gurtler N, Krieger JA, *et al.* Three-Dimensional Electronic Structure of the Type-II Weyl Semimetal WTe₂. *Phys Rev Lett* 2017, **119**(2): 026403.

52. Ultrafast Light-Induced Lifshitz Transition. *arXiv:200304059*.

53. Hübener H, Sentef MA, De Giovannini U, Kemper AF, Rubio A. Creating stable Floquet-Weyl semimetals by laser-driving of 3D Dirac materials. *Nat Commun* 2017, **8**: 13940.”

Comments from Reviewer 2:

The manuscript contains a theoretical analysis of the photoexcitation of the material WTe₂. The focus is on the pump-field polarization dependence of the photoexcited dynamics of the material. It is showed that photoexcitation with a 0.6eV pulse polarized along the (a) and (b) cristallographics axis leads to the excitation of the interlayer shear motion in opposite directions. It is argued that this polarization dependence can be used to tune the position and the number of the Weyl Points (WPs) in the Brillouin Zone. It is therefore proposed that the light-polarization could be used to tune the topological properties of this material.

The idea of using light to control topological properties (namely the position and the number of the WPs) using light stimulation of the interlayer shear mode in WTe₂ is not new. In fact, it has been recently investigated experimentally in Nature 565, 345 (2019) [Ref.37] which is used by the authors to compare part of their theoretical results. The authors provides an extended microscopic description of this phenomenon based on an ab-initio description of the photoexcited dynamics. This study makes the additional observation that, based on a symmetry analysis of the bands close to the WPs, this phenomenon can be controlled using the polarization of the incident light. The polarization dependence of the photoexcitation in WTe₂ is an already documented fact, see for example ARPES data in J. Phys. Condens. Matter 32, 345503 (2020) [Ref. 40]. This paper contains an interesting observation which builds on the combination of the two aspects above. Overall I think that this represents an incremental advancement of the knowledge about the light-control of topological properties of WTe₂. The paper seems scientifically sounds, even if there are points that needs to be addressed in more detail.

Our response: We thank the reviewer for pointing out “this paper contains an interesting observation which builds on the combination of the two aspects above” and “the paper seems scientifically sounds.” The reviewer has some concerns on the

novelty of the present work and the differences from previous works, which might result from misunderstanding the key messages of this work and some misleading impressions on the experimental data in literature. We have addressed all concerns of the reviewer and thoroughly revised the manuscript following the detailed suggestions.

I find that the symmetry analysis which is at the basis of the polarization selection rules is not sufficiently explained. Moreover, except for the plot in Fig.5c, the paper lacks of an in-depth comparison with the experimental observation of Ref. 37. In Ref. 37 the photoinduced shear mode was explicitly reported to be polarization independent (see Fig. 3c in ref. 37). Since the present study points towards a polarization dependence of the shear mode excitation, this aspect needs to be discussed. The paper can be certainly improved, but my opinion is that this work does not meet the criteria for publication of Nature Communication.

Our response: We are sorry for the confusions the reviewer may encounter due to the insufficient explanation and for that the importance and the innovation of this work are not fully recognized by the reviewer. Combined with new revisions and detailed responses to specific comments, the significance of this work is further clarified.

Comments 1: *The symmetry analysis of the selection rules is not sufficiently explained. Bands 0 1 and 2 that seems to play a major role in this. These bands are shown in Fig. 2a, but I find it difficult to recognize these bands in panels b and c where many more bands are plotted. Next to that the selction rules related to the dipole matrix elements $M_{ij} = \langle \Psi_i | X | \Psi_j \rangle$. What are the crystal symmetries that leads to the vanishing of cerntains matrix elements? For example, similar aspects has been recently addressed in the context of the excitonic insulator condidate Ta₂NiSe₅ (see PRR 2, 013236 (2020) and PRL 124, 197601 (2020)).*

Our response: We are sorry for the confusion. In the present work, we propose that the unique wavefunction features around the Weyl node of T_d -WTe₂ lead to a new kind of linear-polarization-dependent photoexcitation for chirality control of Weyl quasiparticles. To demonstrate this point, orbital-projected band structures are displayed for band 0, 1 and 2, as shown in Fig.2 (b) and 2(c). Figure 2(c) shows the magnified band structure around the Weyl node (i.e., the black rectangle region in Fig. 2(b)) with more detailed orbital information, which is important for analyzing the selective excitation. The size of the colored dots represents the weight of the corresponding atomic orbitals. The selection rule is based on the crystal symmetry analysis, similar to that adopted in the previous work [PRL 124, 197601 (2020)].

Detailed explanation is described as follows (SI, Page 3):

“**The optical transition probability rate for an electron that is excited by a photon from the valence band to the conduction band can be described by the Fermi golden rule. Based on time-dependent perturbation theory, the transition probability from an initial state $|\psi_i\rangle$ to a final state $|\psi_f\rangle$ is**

$$\Gamma_{if} = \frac{2\pi}{\hbar} |\langle \psi_f | H' | \psi_i \rangle|^2 \delta(E_f - E_i - \hbar\omega). \quad (\text{R6})$$

The operator for the interaction between the system and the electric field is

$$H' \approx \frac{eA}{m} \vec{\epsilon} \cdot \vec{p}, \quad (\text{R7})$$

where $\vec{\epsilon}$ is the laser polarization vector and $\vec{p} = \frac{\hbar}{i} \frac{\partial}{\partial \vec{r}}$ is the momentum operator.

For the wavefunctions as Bloch waves, we have $\psi_i(\vec{r}, t) = \mu_i(\vec{r}, t) e^{i\vec{k} \cdot \vec{r}}$, and hence, $\Gamma_{if} \propto \left| \langle \psi_f | \vec{\epsilon} \cdot \frac{\partial}{\partial \vec{r}} | \psi_i \rangle \right|^2$. Therefore, the optical transition probability strongly depends on the details of the initial and final state wave functions and the polarization direction of the incident laser pulses. In practice, it is sufficient to analyze the symmetry of the transition dipole moment $M_{if} = \langle \psi_f | \vec{\epsilon} \cdot \hat{r} | \psi_i \rangle$ to determine a selection rule. If the symmetry of M_{if} spans the totally symmetric representation of the point group to which the crystal belongs (i.e., an even

function), then its integral over all space is not zero and the transition is allowed. Otherwise, the transition is forbidden.

The symmetry of the transition moment function is the direct product of the parity of its three components $\Gamma_f \otimes \Gamma_r \otimes \Gamma_i$. The symmetry characteristics of each component can be obtained from standard character tables. For T_d -WTe₂, the character table and the multiplication table are shown as Table S1 and S2, respectively. Let us consider transition from a p_x orbital of the Te atom to a d_{z^2} orbital of the W atom, which have the symmetry of the B₁ and A₁ irreducible representation, respectively. For a laser pulse with polarization direction along the crystallographic a -axis (real-space x -axis), its irreducible representation is B₁. The direct product of the parities of these three components yields an A₁ irreducible symmetry representation (i.e., none of the symmetry operations changes it), therefore, the transition is allowed. However, a laser pulse polarized along the crystallographic b -axis (real-space y -axis) leads to a symmetry characteristic of A₂, which is anti-symmetric under $\sigma_v(xz)$ and $\sigma_v'(yz)$ operations, and therefore, the integral of M_{ij} over all space will be zero and the transition is forbidden. Some possible transition pathways near the Weyl nodes of T_d -WTe₂ are summarized in Table S3.

Table R1. The character table for the C_{2v} symmetry point group

	E	$C_2(z)$	$\sigma_v(xz)$	$\sigma_v'(yz)$	linear, rotations	quadratic
A ₁	1	1	1	1	z	x^2, y^2, z^2
A ₂	1	1	-1	-1	R_z	xy
B ₁	1	-1	1	-1	x, R_y	xz
B ₂	1	-1	-1	1	y, R_x	yz

Table R2. The multiplication table for the C_{2v} symmetry point group

	A ₁	A ₂	B ₁	B ₂
A ₁	A ₁	A ₂	B ₁	B ₂

A₂	A ₂	A ₁	B ₂	B ₁
B₁	B ₁	B ₂	A ₁	A ₂
B₂	B ₂	B ₁	A ₂	A ₁

Table R3. Optical transition selective rules in T_d -WTe₂

Γ_f	Γ_r	Γ_i	$\Gamma_f \otimes \Gamma_r \otimes \Gamma_i$	Transition allowed?
d_{z^2}	x	p_x	$A_1 \otimes B_1 \otimes B_1 = A_1$	Yes
d_{z^2}	x	d_{xz}	$A_1 \otimes B_1 \otimes B_1 = A_1$	Yes
d_{yz}	y	d_{z^2}	$B_2 \otimes B_2 \otimes A_1 = A_1$	Yes
d_{z^2}	y	p_x	$A_1 \otimes B_2 \otimes B_1 = A_2$	No
d_{z^2}	y	d_{xz}	$A_1 \otimes B_2 \otimes B_1 = A_2$	No
d_{yz}	x	d_{z^2}	$B_2 \otimes B_1 \otimes A_1 = A_2$	No

”

In the revised manuscript, we added the following discussions and relevant references in the main text:

- 1) Page 4: **“Based on group theory analysis, the integral of M_{ij} over all space will be zero if the direct product representation $\Gamma_f \otimes \Gamma_r \otimes \Gamma_i$ is not even parity under all symmetry operations of the group³⁸. The space group of T_d -WTe₂ is $P_{mn2_1}(C_{2v})$, with two reflection symmetries (a mirror in the y-z plane and a glide plane in the x-z plane) and a two-fold rotation symmetry C_2 in the unitcell⁶.”**
- 2) Page 5: **“These characteristic transitions are forbidden under the other excitation condition due to the broken of reflection symmetries, i.e., $\langle d_{z^2} | \hat{y} | p_x \rangle = \langle d_{z^2} | \hat{y} | d_{xz} \rangle = \langle d_{yz} | \hat{x} | d_{z^2} \rangle = 0$. (for detailed analysis, see Supplementary Materials).”**
- 3) In supporting information (SI, Page 3), Eqs. (R6) to (R7) and related discussions have been added.
- 4) New references are added:

“38. Damascelli A, Hussain Z, Shen Z-X. Angle-resolved photoemission studies

of the cuprate superconductors. *Rev Mod Phys* 2003, 75(2): 473-541.”

Comments 2: What is the meaning of the states Ψ_i ? Are these Bloch states, or Wannier projection or atomic orbitals? In the latter case, selection rules usually refer to spherical symmetry. How these are extended to the solid?

Our response: We clarify that ψ_i are Bloch states.

In the revised manuscript, we added the following statement in the main text on Page 4:

“According to the dipole selection rule, the transition probability depends on the geometry of the orbitals, i.e., $M_{if} = \langle \psi_f | \hat{r} | \psi_i \rangle$, where M_{ij} is the transition matrix element between the initial Bloch state $|\psi_i\rangle$ and final Bloch state $|\psi_f\rangle$.”

Comments 3: In Ref. 37 the excitation of the interlayer shear mode was first achieved with a pump in the THz regime. What is expected in the theory analysis for a frequency smaller than the 0.6 eV considered here? Also, what is the expected fluence dependence?

Our response: We are grateful for the reviewer’s suggestion requesting in-depth comparison to the experimental observations in Ref. 37, i.e., the polarization dependence of the interlayer shear displacements (Fig. 3c in Ref. 37). The selective excitations originate from the asymmetric distribution of electron orbitals around the Weyl points. Therefore, the induced polarization-anisotropic response of interlayer shear displacement emerge only if the excitation involves transitions between bands forming (or nearing) the Weyl cone. It is photon-energy sensitive, which determines the transition pathways between the initial and final Bloch states. For a frequency much smaller than the 0.6 eV, i.e., in THz regime, the photon energy (e.g., 23 THz \approx 0.095 eV) is too low to promote electrons to higher-lying conduction bands (e.g., band-2), therefore, only intraband excitations or interband transitions between band-1 and band-0 (path ‘LP-a’ in Fig. 2c) is allowed regardless laser polarization.

Therefore, polarization-isotropic interlayer displacement is expected for THz laser irradiation, which is consistent with experimental observations (Fig. 3c in Ref. 37).

To demonstrate the fluence dependence, three laser pulses polarized along the crystallographic b -axis are applied to T_d -WTe₂ with different laser field amplitudes. The corresponding interlayer shear displacements are shown in Fig. R3. It is obvious that with the increase of laser field amplitude, the atomic movements are accelerated and are linearly dependent on the laser fluence.

Figure R3. Averaged displacement of the top layer atoms under various field fluences. The laser duration are the same as those shown in Fig. S1a for all the light pulses.

In the revised manuscript, we have added the following discussion and explanation to the revised text. We hope the revised manuscript is now more clear to the readers.

1) Page 10:

“Selective excitation of WSMs emerge only if the excitation involves transitions between bands forming (or nearing) the Weyl cone, leading to polarization-dependent shear mode. However, when a terahertz laser pulse is applied, the photon energy (e.g., 23 THz \approx 0.095 eV) is too low to promote electrons to higher-lying conduction bands (e.g., band-2), therefore, only intraband excitations and interband transitions between band-1 and band-0

(path ‘LP-a’ in Fig. 2c) is allowed regardless laser polarization. On the other hand, a much higher photon energy (e.g., 1.5 eV) would excite electrons to bands far above the Weyl points, making the process of marginal relevance to Weyl physics (path ② in Fig. 2b). Therefore, polarization-isotropic interlayer displacement is expected in the above two excitation conditions (i.e., THz and visible excitations), both lead to the annihilation of the WPs.”

2) In supporting information (SI, Page 6), Figure R3 and related discussions have been added.

Comments 4: What is the polarization used in the experimental data used to compare the TDDFT results at LP-a polarization?

Our response: The experimental data used to compare the TDDFT results is obtained under the pump pulse excitation polarized at 45° of the crystallographic *a*-axis (Fig. 4e in Ref. 37).

In the revised manuscript, we added the following explanations to make it clearer (Page 19):

“The SHG time trace is measured at pump field polarized at 45° of the crystallographic *a*-axis, the laser pulse strength and wavelength are 10 MV cm⁻¹ and 2.1 μm, respectively.”

Final notes on the revised manuscript:

Based on the above discussions, although the main results of this work build upon the combination of two well-established ideas, i.e., laser controlled topological properties and the polarization dependence of the photoexcitation, the connection between the unique orbital features around the Weyl nodes and the polarization-switchable interlayer displacement is completely a new discovery. The orbital-selective excitation to achieve chirality control is intrinsically different from the well-known chiral selection rules based on spin and circularly polarized light, thus

adding a new knob to the existing rich physics of Weyl points.

We note that the present work is significantly different from the works appeared recently in literature. Most well-known papers on Weyl physics of WTe_2 do not present information on orbitals around the Weyl points (Nature 527, 495 (2015); PRB94,241119 (2016)); a few latest articles indeed performed orbital analysis, however, only on bands far away from the Weyl points, thus are irrelevant to chiral physics (JPCM 32,345503 (2020); JPCM 27, 28540 (2015)). Our paper presents a contribution distinct from that of Ref. 37 (Nature 561, 61 (2019)) in that: i) Our investigation covers a wide range of photoexcitation conditions (photon energy: 0.5-1.5 eV; incident angle: 0-90°), while only THz laser is used in Ref. 37; ii) Therefore the driving mechanisms are different in the two papers: optical excitation versus THz-field-driven low-energy phonons; iii) Most importantly, only restoring of inversion symmetry is presented in Ref. 37 where the pathway is speculative, we present here a complete phase diagram (Fig. 1b) where crystal asymmetry can be either destroyed or enhanced at wish, and the real time dynamics is obtained fully from TDDFT quantum dynamics simulations.

None of these papers mentioned above has in any means implied the potential orbital selectivity to achieve chirality control at the Weyl points, the key point proposed in the present work. Therefore, we believe that this work is a significant contribution and would be of general interest to the broad readership of *Nature Communications*.

In the revised manuscript, we added the following discussions and relevant references to emphasis the importance and the novelty of this work:

1) Page 9:

“The main results of this work established a connection between the unique orbital features around the Weyl nodes and the polarization-switchable interlayer displacement, which is significantly different from the works that appeared recently in literature. Most well-known papers on Weyl physics of WTe_2 do not present information on orbitals around the Weyl points^{6,49}; a few latest articles indeed performed orbital analysis, however, only on bands far

away from the Weyl points, thus are irrelevant to chiral physics⁴⁴. Our investigation covers a wide range of photoexcitation conditions (photon energy: 0.5-1.5 eV; incident angle: 0-90°), and therefore the driving mechanisms, i.e., optical excitation and effective electron-phonon couplings are distinct from the THz-field-driven symmetry switch reported recently³⁷. Most importantly, only restoring of inversion symmetry is presented in Ref. 37 where the pathway is speculative, we present here a complete phase diagram (Fig. 1b) where crystal asymmetry can be either destroyed or enhanced at wish, and the real time dynamics is obtained fully from TDDFT quantum dynamics simulations.”

2) New references are added:

“44. Hein P, Jauernik S, Erk H, Yang L, Qi Y, Sun Y, *et al.* A combined laser-based angle-resolved photoemission spectroscopy and two-photon photoemission spectroscopy study of Td-WTe₂. *J Phys Condens Matter* 2020, **32**(34): 345503.

49. Wang C, Zhang Y, Huang J, Nie S, Liu G, Liang A, *et al.* Observation of Fermi arc and its connection with bulk states in the candidate type-II Weyl semimetal WTe₂. *Phys Rev B* 2016, **94**(24): 241119.”

Finally, we would like to thank all the reviewers for the patience, the time and the helpful suggestions. All the specific comments are responded point-to-point and corresponding revisions are made in the manuscript. Based on that, we believe that the quality and clarity of our work are significantly improved.

Reviewers' Comments:

Reviewer #1:

Remarks to the Author:

The authors have convincingly answered the questions raised in the first round, and I can now recommend publication of the revised manuscript in Nature Communications.

Reviewer #2:

Remarks to the Author:

I have read the author's reply to my comments.

The authors improved their manuscript, especially for what concerns the extended discussions of the symmetries in the supplemental note.

I still suggest the authors to improve Fig. 2, panel c.

There are three bands close to the WP plotted in panel a, and labelled 0, 1 and 2.

In panel c, I can count six/seven bands.

How one can identify bands 0, 1 and 2 out of these six bands?

Maybe it is useful to put some label.

Also I am still confused with the attempt of the authors to compare with the SHG data of ref.37 in Fig. 5c. I am not satisfied by the authors' reply there.

TDDFT data are shown for LP-a polarisation while experimental traces are reported for 45-degrees with respect to a axis polarisation.

Why the two different polarisation should be comparable? This is important since this paper shows that polarisation of light plays a crucial role.

Given that, what is the point of comparing theory and experiments at different polarisations?

Reviewer #3:

Remarks to the Author:

In this paper the authors present a theoretical study of light-induced manipulation of Weyl points in WTe₂ based on ab-initio TDDFT plus Ehrenfest MD numerical simulations.

The authors find that a particular orbital selection rule at play in the proximity of the WP is responsible, when pumped, of a finite charge accumulation in the unit cell that in turn can initiate an interlayer shear motion of the crystal structure. This particular ionic distortion has been studied in a recent experimental paper by Sie et. al [37] in connection with Weyl point annihilation in WTe₂. In contrast with the experimental work where the shear mode was directly excited with THz pulses the mechanism presented in this work allows for a control of the direction of the motion that it can be tuned by the in-plane direction of a linearly polarized NIR field (0.6 eV).

This is a finding that, if confirmed by further studies, would open a new pathway/mechanism to manipulate Weyl nodes and their properties in WTe₂ and the theoretical approach appears solid and predictive enough to warrant an independent experimental validation.

The novelty presented by the results contained in the paper is, in my option, high enough for the standards of Nature Communications.

I have few comments.

1. In Fig S2 a the authors present the time-evolution of the carrier density on the highest valence band. How is this quantity calculated in the simulations with moving ions? The projection over the ground-state KS states with eq (S4) is well defined only when the ions are fixed because the equilibrium basis is no longer a good basis if the geometry changes (the bandstructure can also be very different depending on electron-phonon coupling). In the case of of Fig S1 the projection

seems correctly applied to the system with clamped ions (and also in a relatively short time scale up to 100fs) but the application of eq S4 to a time evolution with moving ions on 800fs is unlikely to provide any meaningful information on the carrier dynamics or other dynamics of the system. The authors should clarify this point and explain what sort of information is contained.

2. I believe the use of the length-gauge formulation for the dipole matrix element, $\langle r \rangle$, like done extensively in the "Unique orbital features around the Weyl nodes" section is not appropriate for infinite systems (exception made when the states are expanded in a Wannier basis), see eg. [Resta, Vanderbilt "Theory of Polarization: A Modern Approach"]. Instead of the matrix elements of the position operator, $r.E$, the authors should take the elements of the momentum operator, $p.A$. I think the selection rules they discuss will not be affected substantially by the change but this aspect should be considered carefully.

3. While the phase diagram of Fig. 1 b is a bit misleading the reader into thinking that the a given field frequency and polarization energy would provide a certain WP node separation from the paper it clearly emerges that the effect of the field would be to initiate a particular oscillatory shear motion. One can thus imagine the WP evolution to take place in an oscillatory fashion around their equilibrium position with the effect of the field parameters being that of affecting the initial "phase" (attractive/repulsive) of the motion. This implies that any experimental observation of the WP dynamics predicted should be transient and locked-in to the frequency of the shear mode. I think the authors should be more explicit about this aspect in the paper and that they should elaborate more on the challenges that this will pose to the experimental observation.

Comments from Reviewer 1:

The authors have convincingly answered the questions raised in the first round, and I can now recommend publication of the revised manuscript in Nature Communications.

Our response: We thank the reviewer for the recommendation of publication in *Nature Communications* without reservation.

Comments from Reviewer 2:

I have read the author's reply to my comments. The authors improved their manuscript, especially for what concerns the extended discussions of the symmetries in the supplemental note.

Our response: We thank the reviewer for pointing out that the manuscript has been improved. We have further revised the manuscript based on the reviewer's constructive suggestions.

Comments 1: *I still suggest the authors to improve Fig. 2, panel c. There are three bands close to the WP plotted in panel a, and labelled 0, 1 and 2. In panel c, I can count six/seven bands. How one can identify bands 0, 1 and 2 out of these six bands? Maybe it is useful to put some label.*

Our response: Many thanks for the suggestion. In the revised manuscript, Fig. 2 is improved by showing only three bands for orbital information in panel (c). The updated version is consistent with what shown in panel (a)-(b) for better illustration. The results of symmetry analysis are unaltered as before.

Fig. 2| Band structure of WTe₂. **a**, Band structure of T_d -WTe₂ in the vicinity of two Weyl nodes. **b**, Band structure along the high-symmetry lines in the Brillouin zone. The radiuses of cyan and yellow dots indicate the weight of W-5d and Te-5p orbitals. The red dot represents the position of WP1. Arrows ① and ② show the carrier transitions that near or far away from the Weyl node. The black rectangle represents the energy range where orbital-selective photoexcitation occurs. **c**, The magnification of the band structure along X- Γ , and with more detailed atomic orbital information. The size of the colored dots represents the weight of the corresponding atomic orbitals. The red arrows represent the transitions with the laser linearly polarized along the a -axis (LP-a excitation) and b -axis (LP-b excitation) with a photon energy of 0.6 eV.

Comments 2: I am still confused with the attempt of the authors to compare with the SHG data of ref.37 in Fig. 5c. I am not satisfied by the authors' reply there. TDDFT data are shown for LP-a polarization while experimental traces are reported for 45-degrees with respect to a axis polarization. Why the two different polarizations should be comparable? This is important since this paper shows that polarization of light plays a crucial role. Given that, what is the point of comparing theory and experiments at different polarizations?

Our response: We are sorry for this confusion. The motivation of the comparison is to demonstrate that the interlayer displacements will establish an oscillating behavior for a longer time scale, which are difficult to be investigated directly in our calculations owing to unaffordable computational cost. We fully agree that the comparison should be performed with the same polarization.

Following the same methods as introduced in the main text, new TDDFT-MD

simulations are performed with the laser polarization being 45° off the crystallographic a -axis (Fig.R1a). The corresponding laser-induced interlayer shear displacement is shown in Fig.R1b. It is clear that the shear motion leads to the restoring of inversion symmetry, which is consistent with the experimental observations³⁷. We notice that the atomic displacements are slower than that under the LP-a and LP-b excitations ($\approx 0.076 \text{ \AA/ps}$), which might be attributed to the fact that the accumulative effect of photoexcited carriers is partially cancelled out by its two polarization components. The non-centrosymmetric degree (NCD) of WTe_2 under this polarization condition ($E = 2.8 \text{ MV/cm}$) matches well with the experimentally detected SHG signal ($E = 3.1 \text{ MV/cm}$), as shown in Fig. R1c. Based on that, we expect that the interlayer displacements might exhibit an oscillating behavior in later stages.

Fig. R1 | Interlayer shear displacement with the laser polarization being 45° off the crystallographic a -axis. **a**, Schematic illustration of the calculation setup with the laser pulse are linearly polarized 45° off the crystallographic a -axis ($\theta = 45^\circ$). The other laser parameters are same as those shown in Fig. S1a. **b**, Interlayer shear displacement under the excitation. **c**, Comparison of non-centrosymmetric degree (NCD) between TDDFT result and the experimental detected SHG signal taken from Ref. [37]. In both two cases, the pump fields are polarized at 45° off the crystallographic a -axis. The SHG time trace is measured at the laser pulse strength and wavelength are 3.1 MV/cm and 2.1 \mu m , respectively.

In the revised manuscript, we have added the following discussions and explanation to the revised text. We hope the revised manuscript is now clearer to the readers.

1) Page 8:

“Limited by the unprecedented computational cost of first-principles excited-state dynamic simulations, the time duration we could simulate is less than 1 ps, which only describes the early stage of the photoinduced phase transition. It is reported that under moderate laser intensities, the interlayer displacements will oscillate with the frequency of the shear mode³⁷. Qualitative prediction of the TDDFT lattice dynamics in a longer time scale can be achieved via comparing the time evolution of non-centrosymmetric degree (NCD) of WTe₂ between our results with experimental data. It can be verified using time-resolved second-harmonic generation (SHG) technique³⁷ or by monitoring the interlayer shear displacements (Supplementary Fig. 6). Figure 5c shows the normalized NCD when the pump fields are polarized at 45° off the crystallographic *a*-axis, for the initial *T_d* phase, NCD is 1; for centrosymmetric phase, NCD is 0. The time-evolution of NCD match well with the experimentally detected SHG signal, justifying the reliability of the present approach. In Supplementary Fig. 6(d), we show that after *t* = 600 fs, the NCD of WTe₂ tends to return to its initial state under both LP-a and LP-b excitations. Based on that, we expect the interlayer displacements might exhibit an oscillatory behavior in later stages.”

2) Page 24: Fig.5c and the corresponding caption are updated as what we shown here (Fig. R1c).

3) In supporting information (SI, Page 11), Fig.R1b and related discussions have been added.

Comments from Reviewer 3:

In this paper the authors present a theoretical study of light-induced manipulation of Weyl points in WTe_2 based on ab-initio TDDFT plus Ehrenfest MD numerical simulations. The authors find that a particular orbital selection rule at play in the proximity of the WP is responsible, when pumped, of a finite charge accumulation in the unit cell that in turn can initiate an interlayer sheer motion of the crystal structure. This particular ionic distortion has been studied in a recent experimental paper by Sie et. al [37] in connection with Weyl point annihilation in WTe_2 . In contrast with the experimental work where the sheer mode was directly excited with THz pulses the mechanism presented in this work allows for a control of the direction of the motion that it can be tuned by the in-plane direction of a linearly polarized NIR field (0.6 eV). This is a finding that, if confirmed by further studies, would open an new pathway/mechanism to manipulate Weyl nodes and their properties in WTe_2 and the theoretical approach appears solid and predictive enough to warrant an independent experimental validation. The novelty presented by the results contained in the paper is, in my option, high enough for the standards of Nature Communications.

Our response: We are grateful to the reviewer for his/her praise and for pointing out the novelty of our paper “*This is a finding that, if confirmed by further studies, would open an new pathway/mechanism to manipulate Weyl nodes and their properties in WTe_2 and the theoretical approach appears solid and predictive enough to warrant an independent experimental validation.*” and “*The novelty presented by the results contained in the paper is, in my option, high enough for the standards of Nature Communications.*”

The reviewers’ comments and suggestions are very helpful for us to further improve the presentation and quality of our work. Following his/her suggestions, we have addressed the concerns raised by the reviewer.

Comments 1: In Fig S2 the authors present the time-evolution of the carrier density on the highest valence band. How is this quantity calculated in the simulations with moving ions? The projection over the ground-state KS states with eq (S4) is well defined only when the ions are fixed because the equilibrium basis is no longer a good basis if the geometry changes (the band structure can also be very different depending on electron-phonon coupling). In the case of Fig S1 the projection seems correctly applied to the system with clamped ions (and also in a relatively short time scale up to 100fs) but the application of eq S4 to a time evolution with moving ions on 800fs is unlikely to provide any meaningful information on the carrier dynamics or other dynamics of the system. The authors should clarify this point and explain what sort of information is contained.

Our response: Many thanks for the comments. We are sorry for that some detailed methods were not sufficiently explained. To be short, the time-evolving population is obtained by **projecting TDKS to the adiabatic KS states at each ionic configuration**, not to the initial KS states as in clamped-ion simulations. We elaborate in more detail below.

In our calculations, the time-dependent coupled electron-ion motion can be computed by propagating the Kohn-Sham equations,

$$i \frac{\partial}{\partial t} \psi_i(\vec{r}, t) = \left[\frac{1}{2m} \left(\vec{p} - \frac{e}{c} \vec{A} \right)^2 + V(\vec{r}, t) \right] \psi_i(\vec{r}, t). \quad (\text{R1})$$

Due to the fact that the ions are much heavier than electrons by at least three orders of magnitude, the nuclear positions are updated following the Newton's second law,

$$M_I \frac{d^2 \mathbf{R}_I}{dt^2} = \sum_i \left\langle \psi_i \left| \nabla_{\mathbf{R}_I} \left(\frac{1}{2m} \left(\vec{p} - \frac{e}{c} \vec{A} \right)^2 + V(\mathbf{r}, t) \right) \right| \psi_i \right\rangle, \quad (\text{R2})$$

where M_I and \mathbf{R}_I are the mass and position of the I^{th} ion, respectively.

The TDKS orbitals $\psi_{n,\mathbf{k}}(\vec{r}, t)$ can be expressed by the combination of adiabatic basis $\{\varphi_{n,\mathbf{k}}(\vec{r}, t)\}$

$$|\psi_{n,\mathbf{k}}(\vec{r}, t)\rangle = \sum_{n'} c_{nn'k}(t) |\varphi_{n',\mathbf{k}}(\vec{r}, t)\rangle, \quad (\text{R3})$$

where n and n' denote the band index, \mathbf{k} is the reciprocal momentum index, $c_{nn'\mathbf{k}}(t)$ is the time dependent coefficients. The adiabatic basis $\varphi_{n,\mathbf{k}}(\vec{r}, t)$ are solved by diagonalizing the Hamiltonian

$$H_{n,\mathbf{k}}(\vec{r}, t)|\varphi_{n,\mathbf{k}}(\vec{r}, t)\rangle = \varepsilon_{n,\mathbf{k}}(t)|\varphi_{n,\mathbf{k}}(\vec{r}, t)\rangle, \quad (\text{R4})$$

where $\varepsilon_{n,\mathbf{k}}$ is the eigenvalue.

During dynamic simulations, the evolving time step is set to 0.05 fs for both electrons and ions in a micro-canonical ensemble. Based on the ground-state potential energy surface of a certain atomic configuration $R(t)$, the adiabatic basis $\varphi_{n,\mathbf{k}}(\vec{r}, t)$ is calculated on the fly at each ionic step. Therefore, the projection of the time-evolving wavefunctions ($|\psi_{n,\mathbf{k}}(t)\rangle$) on the basis of the adiabatic Kohn-Sham orbitals ($|\varphi_{n',\mathbf{k}}\rangle$) represent the state-to-state transition probabilities

$$P_{nn'\mathbf{k}}(t) = |c_{nn'\mathbf{k}}(t)|^2 = |\langle\varphi_{n',\mathbf{k}}(t)|\psi_{n,\mathbf{k}}(t)\rangle|^2. \quad (\text{R5})$$

The population $q_{n\mathbf{k}}$ of adiabatic state $n\mathbf{k}$ is thus projected from TDKS orbitals as

$$q_{n\mathbf{k}} = \sum_{n' \in n_{\mathbf{k},occ}} q_{n'\mathbf{k}} P_{nn'\mathbf{k}}, \quad (\text{R6})$$

where $n_{\mathbf{k},occ}$ is occupied state at \mathbf{k} point.

The time-evolution of the carrier density on the highest valence band that shown in Fig. S2a can be defined as

$$q_n = \frac{1}{N_{\mathbf{k}}} \sum_{\mathbf{k}} \sum_{n' \in n_{\mathbf{k},occ}} q_{n'\mathbf{k}} P_{nn'\mathbf{k}}, \quad (\text{R7})$$

where $N_{\mathbf{k}}$ is the total number of the \mathbf{k} -points used to sample the Brillouin zone.

The methods introduced above can be applied to investigate dynamic processes with or without ionic movement, while with the clamped ions, the adiabatic basis $\varphi_{n,\mathbf{k}}(\vec{r}, t)$ is fixed.

To make it clearer, we have added the following text in the revised Supplementary:

Page 2:

“As the ions are much heavier than electrons by at least three orders of magnitude, the nuclear positions are updated following the Newton’s second law,

$$M_I \frac{d^2 \mathbf{R}_I}{dt^2} = \sum_i \left\langle \psi_i \left| \nabla_{\mathbf{R}_I} \left(\frac{1}{2m} \left(\vec{p} - \frac{e}{c} \vec{A} \right)^2 + V(\mathbf{r}, t) \right) \right| \psi_i \right\rangle, \quad (\text{S4})$$

where M_I and \mathbf{R}_I are the mass and position of the I^{th} ion, respectively. Equation (S2) and (S4) represents the time-dependent coupled electron-ion motion. The time-dependent Kohn-Sham equations of electrons and the Newtonian motion of ions are solved simultaneously, with ionic forces along the classical trajectory evaluated through the Ehrenfest theorem.

The TDKS orbitals $\psi_{n,\mathbf{k}}(\vec{r}, t)$ can be expressed by the combination of adiabatic basis $\{\varphi_{n,\mathbf{k}}(\vec{r}, t)\}$

$$|\psi_{n,\mathbf{k}}(\vec{r}, t)\rangle = \sum_{n'} c_{nn'\mathbf{k}}(t) |\varphi_{n',\mathbf{k}}(\vec{r}, t)\rangle, \quad (\text{S5})$$

where n and n' denote the band index, \mathbf{k} is the reciprocal momentum index, $c_{nn'\mathbf{k}}(t)$ is the time dependent coefficients. The adiabatic basis $\varphi_{n,\mathbf{k}}(\vec{r}, t)$ are solved by diagonalizing the Hamiltonian

$$H_{n,\mathbf{k}}(\vec{r}, t) |\varphi_{n,\mathbf{k}}(\vec{r}, t)\rangle = \varepsilon_{n,\mathbf{k}}(t) |\varphi_{n,\mathbf{k}}(\vec{r}, t)\rangle \quad (\text{S6})$$

where $\varepsilon_{n,\mathbf{k}}$ is the eigenvalue.

During the lattice dynamics, based on the ground-state potential energy surface of a certain atomic configuration $R(t)$, the adiabatic basis $\varphi_{n,\mathbf{k}}(\vec{r}, t)$ is calculated on the fly at each ionic step. Therefore, the projection of the time-evolved wavefunctions ($|\psi_{n,\mathbf{k}}(t)\rangle$) on the basis of the adiabatic Kohn-Sham orbitals ($|\varphi_{n',\mathbf{k}}\rangle$) represent the state-to-state transition probabilities

$$P_{nn'\mathbf{k}}(t) = |c_{nn'\mathbf{k}}(t)|^2 = |\langle \varphi_{n',\mathbf{k}}(t) | \psi_{n,\mathbf{k}}(t) \rangle|^2. \quad (\text{S7})$$

The population q_n of band n is thus projected from TDKS orbitals as

$$q_n(t) = \frac{1}{N_{\mathbf{k}}} \sum_{\mathbf{k}} \sum_{n' \in n_{\mathbf{k}, \text{occ}}} q_{n'\mathbf{k}}(t) P_{nn'\mathbf{k}}(t), \quad (\text{S8})$$

where $n_{\mathbf{k},occ}$ is occupied state at \mathbf{k} point and $N_{\mathbf{k}}$ is the total number of the \mathbf{k} -points used to sample the Brillouin zone.

The dynamic of the excited electrons are calculated by

$$\Delta n_e(t) = N_e - \sum_n^{VB} q_n(t), \quad (S9)$$

where all valence-band (VB) electrons are summed up and N_e is the total number of electrons.”

Comments 2: I believe the use of the length-gauge formulation for the dipole matrix element, $\langle r \rangle$, like done extensively in the "Unique orbital features around the Weyl nodes" section is not appropriate for infinite systems (exception made when the states are expanded in a Wannier basis), see eg. [Resta, Vanderbilt "Theory of Polarization: A Modern Approach"]. Instead of the matrix elements of the position operator, $r.E$, the authors should take the elements of the momentum operator, $p.A$. I think the selection rules they discuss will not be affected substantially by the change but this aspect should be considered carefully.

Our response: We fully agree with the reviewer. Based on the reviewer’s suggestion, we have updated the following discussions on Page 4 of the main text:

“According to the dipole selection rule, the transition probability depends on the geometry of the orbitals, i.e., $M_{if} = \langle \psi_f | \vec{\mathcal{D}} \cdot \vec{p} | \psi_i \rangle = \langle \psi_f | \hat{S} | \psi_i \rangle$, where M_{if} is the transition matrix element between the initial Bloch state $|\psi_i\rangle$ and final Bloch state $|\psi_f\rangle$, \vec{p} is the electronic momentum operator, $\vec{\mathcal{D}}$ is the electromagnetic vector potential and $\hat{S} = \vec{\mathcal{D}} \cdot \vec{p}$.”

“We find that $\langle d_{z^2} | S_x | p_x \rangle$ and $\langle d_{z^2} | S_x | d_{xz} \rangle$ are two dominant transition pathways along W- Γ under LP-a excitation, whereas $\langle d_{yz} | S_y | d_{z^2} \rangle$ is the most important pathway along W-X under LP-b excitation (Note crystallographic a -axis and b -axis correspond to the real-space x -axis and y -axis, respectively.) (Fig. 2c).

These characteristic transitions are forbidden under the other excitation condition due to the broken of reflection symmetries, i.e., $\langle d_{z^2} | S_y | p_x \rangle = \langle d_{z^2} | S_y | d_{xz} \rangle = \langle d_{yz} | S_x | d_{z^2} \rangle = 0.$ "

Comments 3: *While the phase diagram of Fig. 1b is a bit misleading the reader into thinking that a given field frequency and polarization energy would provide a certain WP node separation. From the paper it clearly emerges that the effect of the field would be to initiate a particular oscillatory shear motion. One can thus imagine the WP evolution to take place in an oscillatory fashion around their equilibrium position with the effect of the field parameters being that of affecting the initial "phase" (attractive/repulsive) of the motion. This implies that any experimental observation of the WP dynamics predicted should be transient and locked-in to the frequency of the sheer mode. I think the authors should be more explicit about this aspect in the paper and that they should elaborate more on the challenges that this will pose to the experimental observation.*

Our response: We thank the reviewer for pointing out this important aspect. We agree with the reviewer that Fig. 1b shows the initial stages of WP node separation in response to laser frequency and polarization. Based on our simulations, the interlayer displacements and the induced WPs evolution will have an oscillatory fashion around their equilibrium positions (Fig. 5c). Both the polarization and the photon energy of laser pulse will determine the initial phase of the motion. Therefore, the experimental observation requires high temporal resolution, which at least shorter than the period of the sheer mode. Apart from the time-resolved second-harmonic-generation (SHG) or electron diffraction (UED) spectroscopies, which were adopted by Sie *et. al* [37] and Zhang *et. al* [42], time- and angle-resolved photoemission spectroscopy (trARPES) is a potential candidate to achieve this goal. However, direct measurement of the dynamic WPs separation requires sufficient energy and angular resolution to distinguish the terminal of Fermi arc, which might be the major challenges for the experimental observation.

In the revised manuscript, we added the following discussions and relevant references in the main text:

1) We changed the caption of Fig. 1b into “Phase diagram of laser-driven T_d -WTe₂ topological phase transition depending on photon energy $\hbar\omega$ and incident angle θ at the initial stages of the photoexcitation.”

2) New discussions were added on Page 11:

“Based on our simulations, the switchable interlayer displacements and the induced WPs evolution will establish an oscillatory fashion around their equilibrium positions, as shown in Fig.5c. The experimental observation of above dynamics should be transient and locked-in to the frequency of the sheer mode, and therefore, ultrafast techniques with high temporal resolution (~ 300 fs) are needed. Apart from ultrafast pump-probe and time-resolved second-harmonic-generation (SHG) or electron diffraction (UED) spectroscopies^{37,42}, one potential candidate is time- and angle-resolved photoemission spectroscopy (trARPES). It has a demonstrated advantage in investigating the structure of arcs, connectivity of electron and hole pockets and WPs positions. However, direct measurement of the dynamic WPs separation requires sufficient energy and angular resolution to distinguish the terminal of Fermi arc^{43,54,55}, which might be the major challenges for the experimental observation.”

3) New references were added:

54. Crepaldi A, Autès G, Gatti G, Roth S, Sterzi A, Manzoni G, *et al.* Enhanced ultrafast relaxation rate in the Weyl semimetal phase of MoTe₂ measured by time- and angle-resolved photoelectron spectroscopy. *Phys Rev B* 2017, **96**(24).

55. Huang L, McCormick TM, Ochi M, Zhao Z, Suzuki MT, Arita R, *et al.* Spectroscopic evidence for a type II Weyl semimetallic state in MoTe₂. *Nat Mater* 2016, **15**(11): 1155-1160.”

Finally, we would like to thank all the reviewers for the helpful suggestions, the patience, the time and the kind recommendation.

Reviewers' Comments:

Reviewer #2:

Remarks to the Author:

The authors made a further revision of their manuscript taking into account my comments. I appreciated the way they addressed my comments and especially the revision of the comparison of the experimental and theoretical results.

The results now look solid and worth of publication.

Reviewer #3:

Remarks to the Author:

The authors also clearly replied to all my comments to satisfaction and amended the text accordingly.

I have no further comments and recommend the revised version for publication in Nature Communications.